# BASALT refines binning from metagenomic data and increases resolution of genome-resolved metagenomic analysis

Zhiguang Qiu [1,2,22], Li Yuan[2,3,4], Chun-Ang Lian [1,2], Bin Lin[3], Jie Chen[2,3,4], Rong Mu[1], Xuejiao Qiao[1], Liyu Zhang[1], Zheng Xu[5,6], Lu Fan [7], Yunzeng Zhang[8], Shanquan Wang [9], Junyi Li[10], Huiluo Cao[11], Bing Li[12], Baowei Chen[13], Chi Song [14,15], Yongxin Liu [16], Lili Shi[2,17], Yonghong Tian [2,3,4], Jinren Ni [1,18], Tong Zhang [19], Jizhong Zhou [20], Wei-Qin Zhuang[21] & Ke Yu [1,2,22] ✉

Metagenomic binning is an essential technique for genome-resolved characterization of uncultured microorganisms in various ecosystems but hampered by the low efficiency of binning tools in adequately recovering metagenome-assembled genomes (MAGs). Here, we introduce BASALT (Binning Across a Series of Assemblies Toolkit) for binning and refinement of short- and long-read sequencing data. BASALT employs multiple binners with multiple thresholds to produce initial bins, then utilizes neural networks to identify core sequences to remove redundant bins and refine non-redundant bins. Using the same assemblies generated from Critical Assessment of Metagenome Interpretation (CAMI) datasets, BASALT produces up to twice as many MAGs as VAMB, DASTool, or metaWRAP. Processing assemblies from a lake sediment dataset, BASALT produces ~30% more MAGs than metaWRAP, including 21 unique class-level prokaryotic lineages. Functional annotations reveal that BASALT can retrieve 47.6% more non-redundant opening-reading frames than metaWRAP. These results highlight the robust handling of metagenomic sequencing data of BASALT.

Genome-resolved metagenomic analysis has led to major advances in the identification of uncultured microorganisms[1-3]. Several large initiatives have been launched to investigate the scope of microbial diversity on planet Earth, such as the Earth Microbiome Project (EMP), Marine Microbiome Initiative (https://imos.org.au/), and the Urban Microbiome Initiative (https://www.humicity.org), which archived petabase-scale metagenomic data available publicly. For example, the EMP has generated more than 52,000 species level Metagenomic Assembled Genomes (MAGs)[4,5] from human[6,7], freshwater[8], marine[9,10], engineered environment[11,12], and soil[13,14] metagenomic data. However, these findings only scratch the proverbial surface of the microbial diversity likely to occur in nature because MAGs with high completeness and low contamination (hereafter referred to as high-quality MAGs) remain challenging to assemble from environmental samples.

However, this process is nevertheless critical for experimental advances, such as the isolation of uncultivated microorganisms, dissecting novel metabolic or signaling pathways, quantitative determination of metabolic capacity, and decoding microbial interactions[15-20].

Binning is an essential step in genome-resolved metagenomic analysis in which assembled contigs originating from the same source population are clustered based on coverage, contig edges, or tetra nucleotide frequencies (TNFs). However, mis-clustering of contigs into a single bin, erroneously separating contigs from one genome into multiple bins, or mis-sorting multiple genomes into shared bins due to overlapping genomic sequence can result in redundant, artificial, or contaminated bins that interfere with the accuracy of downstream analyses[21-23]. In light of these issues, many new binning[24-27] and refinement tools[28,29] have been developed to

improve resolution in high-diversity environmental sequencing data[30,31]. However, refinement methods based on the reassociation of clustered sequences of single assemblies only use ~30% of sequencing reads that can meet reassociation criteria, effectively further reducing the reads usage capacity (thereafter binning efficiency) for high-diversity environmental samples, such as soils[4,32]. To improve the efficiency of data utilization, short-read sequences (SRS) and long-read sequences (LRS) data can be combined[33,34] in hybrid assembly exercises[35]. However, no tool is currently capable of integrating both SRS and LRS data in the post-binning calibration of assembled bins and complementation of genome gaps (i.e., refinement), which could potentially maximize the number of allowable reads while improving the quality of recovered genomes.

Here, we introduce BASALT, a toolkit for efficient and high-resolution recovery of nonredundant bins from short- and long-read metagenomic sequencing datasets. BASALT can also be used to refine bins from multiple datasets based on neural network, correlation coefficients, and reiteration algorithms to maximize binning efficiency. In a proof-of-concept demonstration with CAMI benchmarking datasets[31], binning with BASALT recovers more and higher-quality MAGs than other leading toolkits. Subsequent analysis of short-read metagenomic sequencing data from sediment samples of Aiding Lake, a previously uncharacterized inland saline lake in the northwest of Xinjiang Province, China, recovers ~30% more MAGs than metaWRAP (Supplementary Fig. 1). BASALT is available as an open-source Python program on GitHub (https://github.com/EMBL-PKU/BASALT).

## Results

### Overview of the BASALT workflow
BASALT is a binning–post-binning refinement tool that reconstructs nonredundant and high-quality Metagenomic Assembled Genomes (MAGs) from metagenomic sequencing data by conducting automatic mapping, binning, and post-binning refinements. To efficiently increase the reads usage capacity and number of high-quality MAGs, the Automated Binning Module performs the initial steps, using either multiple single assemblies (SA) or/and co-assemblies (CA)[36] generated from short-read sequences (SRS) or a mix of SRS and long-read sequences (LRS) as input. For this function, multiple binners are applied, each at multiple thresholds, to increase the number of bin outputs. Second, the Bin Selection Module identifies core sequences from corresponding bins. This module was developed incorporating the concept of coverage correlation coefficient (CCC), which uses an interquartile range (IQR)-based algorithm to calculate contig coverage. These core sequences are further evaluated by the neural network to facilitate the identification and removal of redundant bins generated in the first step. Core sequences are further implemented in the Refinement Module to remove outlier sequences and retrieve un-binned sequences, including multi-copy genes from assembled contigs pool. This module, which also relies on the tetranucleotide frequency (TNF) of each bin, is designed to improve bin completeness and purity. In order to ensure the highest possible MAG quality, the BASALT Gap Filling module implements restrained Overlap–Layout–Consensus (rOLC), accompanied by a de Brujin graph algorithm, to reassemble genomes[37]. The workflow and detailed methods are illustrated in Fig. 1 and explained in detail in "Methods".

Using a high-performance computer workstation (AMD Ryzen Threadripper 3970X @ 3.7 GHz, 32 cores, and 256 GB memory), hybrid assembly of the CAMI-high dataset (SRS + LRS) using OPERA-MS[34] required 35.1 h to complete without polishing, but took another 187.4 h for polishing. Notably, long-read polishing in the hybrid assembly process needed a significant amount of time, and hence subsequent analyses were performed without long-read polishing by OPERA-MS. Using unpolished assemblies, BASALT spent 12.7 h (real time) to finish the Automated Binning and Bin Selecting Modules, 7.8 h to finish the Refinement Module, and 20.8 h to finish Gap Filling module (Supplementary Fig. 2A).

### BASALT generates high-quality MAGs from a benchmarking dataset
To evaluate the binning efficiency of BASALT, Critical Assessment of Metagenome Interpretation (CAMI)[31] that comprises synthetic microbial genome data was selected as benchmarking datasets. Using OPERA-MS[34] to generate hybrid assemblies from the CAMI dataset (CAMI-high, 596 genomes), BASALT recovered 392 nonredundant bins spanning 596 benchmarking genomes (Supplementary Data 1). Using an in-house script (see "Methods") based on average nucleotide identity (ANI) to calculate bin completeness and contamination for comparison with benchmark genomes, BASALT recovered 62.2% of bins (371/596) at baseline quality (completeness ≥35 and contamination ≤20) compared to the gold standard genomes, including 89.8% (333/371) of metagenome-assembled genomes (thereafter MAGs) that met MIMAG standards (completeness−5 * contamination ≥50, thereafter quality ≥50)[38,39] in the final output (Fig. 2a–c).

Assessment of non-coding RNAs (ncRNAs, i.e., rRNA and tRNA) in the recovered bins showed that, among the 371 bins, 5 S, 16 S, and 23 S rRNA reads were found in 91.5%, 59.4% and 74.6% of bins, respectively, whereas 24.5% of bins contained all three types of rRNA sequence (Fig. 2d). In the detection of tRNA genes encoding 20 standard amino acids, 368 of the recovered bins contained tRNA reads, with an average of 18.32 ± 0.12 standard amino acids per bin, including 94% of bins that had ≥15 amino acids (Fig. 2e).

To test the effectiveness of each module in post-binning steps, we assessed the completeness, contamination, and quality of each recovered bins processed by the Bin Selection, Refinement and Gap Filling modules, respectively. Overall, an average increase of 5.26% was found in completeness, while contamination was reduced by 3.76% (a 13.7 overall increase in quality) after refinement, compared with that after Bin Selection. These results indicated that the core sequence extraction method was effective in both outlier removal and Sequence Retrieval steps (Fig. 2f). Comparison of outputs between the Refinement and Gap-filling modules showed a 2.92% average increase in completeness and 3.06% reduction in contamination (overall 13.28 quality increase) after Gap Filling. This finding suggested that the combined rOLC plus de Brujin graph reassembly process was robust, consistent with bin optimization (Fig. 2f). Following the Refinement and Gap Filling modules, the number of high-quality bins was also obviously increased (completeness ≥90 and contamination ≤5) (Fig. 2a, c), suggesting that each BASALT module could generate more high-quality genomes than the previous module. Collectively, these results indicated the high quality of the recovered bins and the efficient recovery of genomes by BASALT.

### BASALT recovers more and higher-quality MAGs from Benchmarking datasets than other binning tools
To further assess the performance of BASALT in assembling genomes from metagenomic data, we used the CAMI benchmarking dataset to compare the outputs of BASALT with two binning pipelines: DASTool[40] and metaWRAP[29], and a recently developed binner VAMB[27]. Overall, a total of 352 MAGs were obtained by all four toolkits together, 168 of which were shared by all four platforms. By contrast, BASALT recovered 69 MAGs that were absent among those generated by the other tools (Fig. 3a). In addition, a comparison of shared MAGs showed that bins produced by BASALT had significantly higher completeness and lower contamination than the same bin produced by the other tools, resulting in significantly higher quality (Kruskal–Wallis test, $P < 10^{-7}$, Fig. 3b and Supplementary Data 2). Pairwise comparison of MAGs delta values, which indicate bin quality, between BASALT and other tools further revealed that MAGs generated by BASALT had ~9.6-, 14.6-, and 6.1-fold higher delta values than the corresponding MAGs produced by VAMB, DASTool, or metaWRAP, respectively (Fig. 3c and

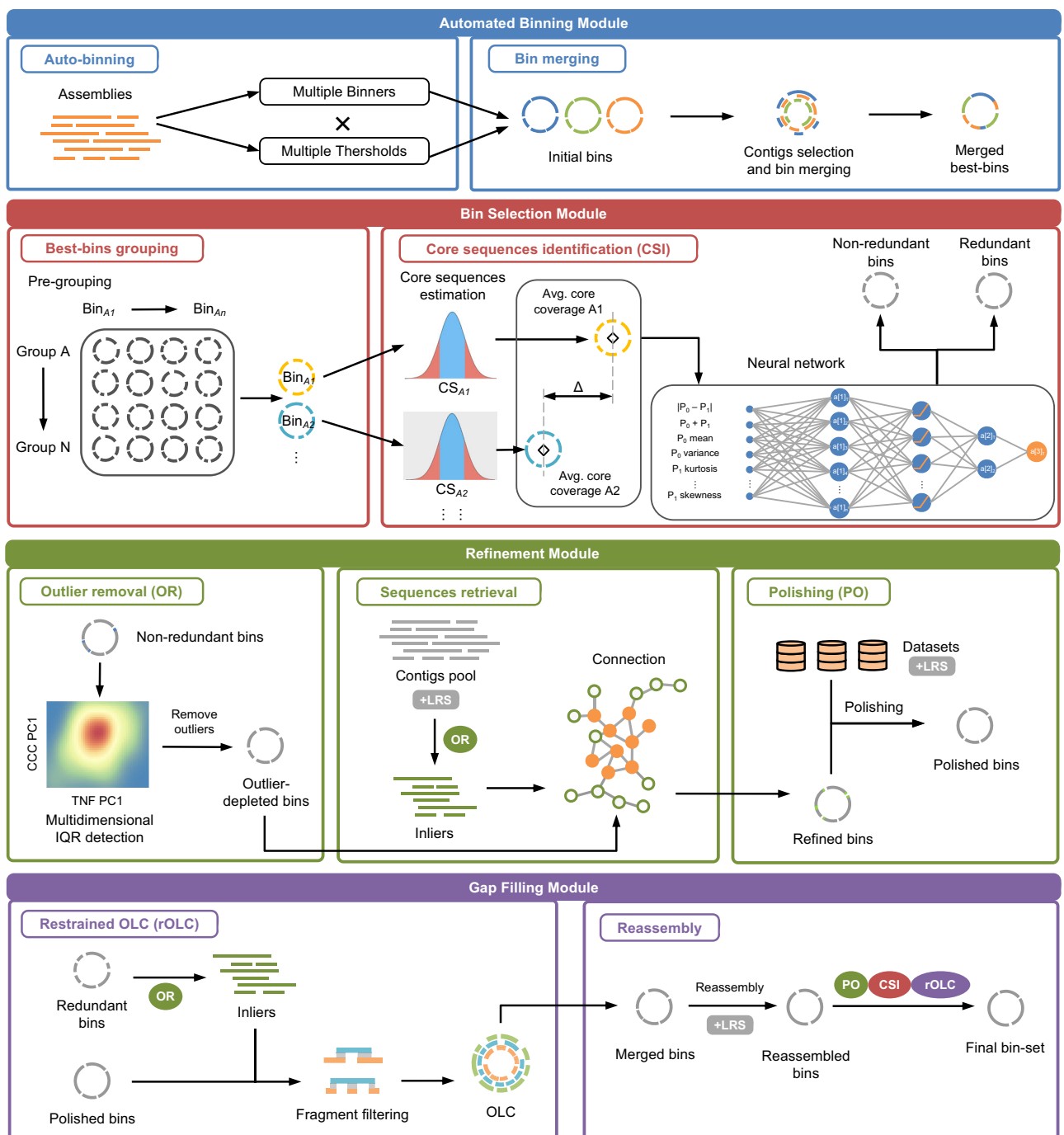

**Fig. 1 | BASALT workflow for assembly, binning, and refinement of short-read sequencing (SRS) and long-read sequencing (LRS) data.** BASALT consists of four modules: Automated Binning, Bin Selection, Refinement, and Gap Filling. First, assembled contigs are sorted into bins by several binning tools, each with multiple thresholds, to create the initial binsets. Bins with similar contigs are merged into hybrid bins by identification of contig IDs. Hybrid binsets are then grouped based on average nucleotide identity (ANI), after which Core sequence identification is used to compare inliers from each pair of bins within groups identified by coverage estimation. Redundant bins are then identified using a neural network algorithm, and nonredundant bins are kept for subsequent Outlier removal. In the refinement module, tetranucleotide frequency (TNF) and coverage correlation coefficient (CCC) are combined for Multidimensional internal quartile range (IQR) detection to identify outlier sequences. Then, a sequence retrieval step connects and reiterates un-binned inliers from the SR and LR contig pools via pair-end (PE) or long-read tracking, resulting in refined bins. Reads that successfully mapped to the refined bins are further polished to generate polished bins. rOLC is then conducted by overlapping refined/polished bins with corresponding redundant bins before reassembly is conducted with both SRS and LRS. Reassembled bins are further polished, followed by another round of rOLC to produce the final bin-set. Blue frame: automated binning module; red frame: bin selection module; green frame: refinement module; purple frame: gap-filling module. +LRS indicates that long-read sequencing data can be used at this step. CSI core sequence identification, OR outlier removal program, rOLC restrained overlap–layout–consensus program.

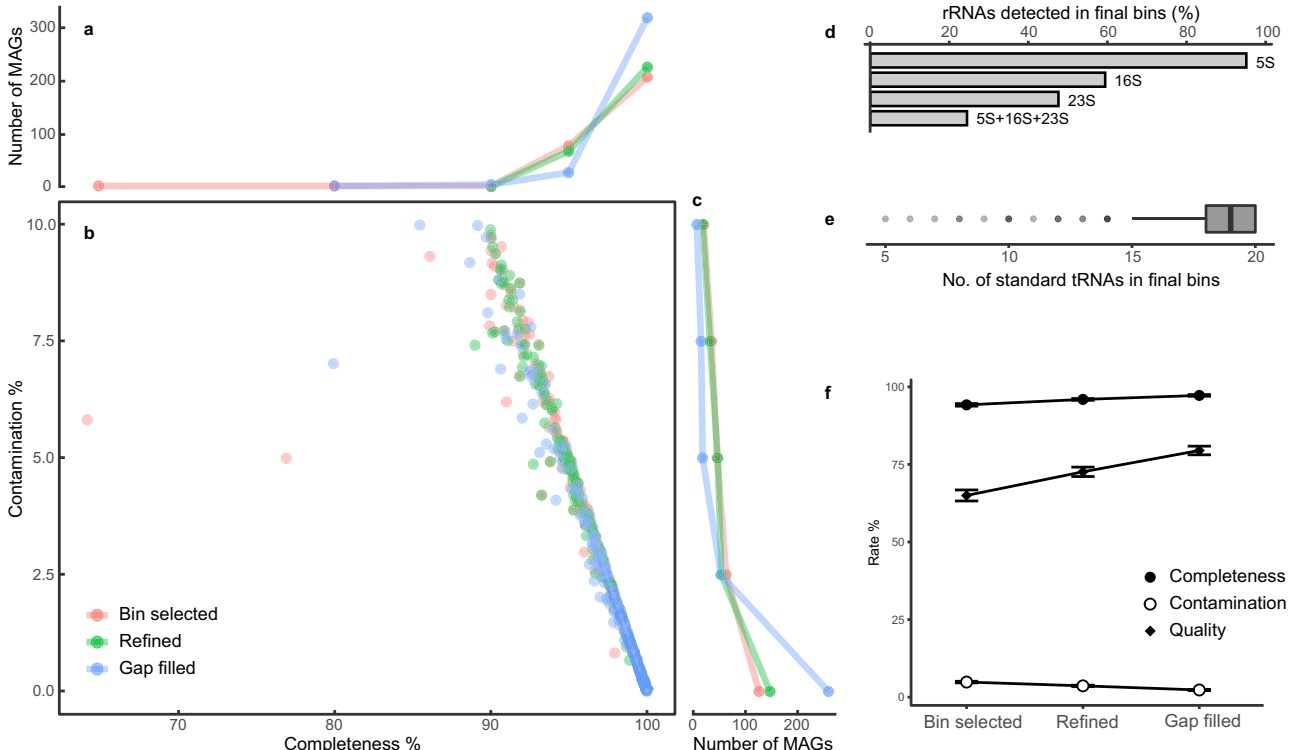

**Fig. 2 | Bin recovery from the CAMI-high dataset using BASALT.** Contigs assembled by Opera-MS were processed with BASALT using default parameters. **a**–**c** Summary of MAGs with completeness ≥50 and contamination ≤10. The number of MAGs are indicated on side bars at the top/right of the main figure. **d** Percentage of rRNAs detected in bins after Gap Filling. **e** Summary of tRNAs in the processed bins after Gap Filling. The boxplot shows the distribution of data, the central dot in the box represents the median, the box bounds represent the 25th and 75th percentiles, and whiskers represent the minima to maxima values. **f** Completeness, contamination and quality rates after processing with Bin Selection (Bin selected, $n = 290$), Refinement (Refined, $n = 298$), and Gap Filling (Gap-filled, $n = 352$) modules. Data are presented as means ± SD from MAGs retrieved at each step.

Supplementary Data 2). In addition, BASALT analyses produced 54.8%, 52.8%, and 26.1% more MAGs than VAMB, DASTool, and metaWRAP, respectively (Fig. 3d and Supplementary Data 2), due in large part to the emphasis on bin refinement in BASALT. Notably, this refinement feature not only led to the recovery of more total MAGs but also more high-quality MAGs, with BASALT yielding 128%, 259%, and 102% more genomes with quality scores ≥90 than VAMB, DASTool, and metaWRAP, respectively (Fig. 3d).

We assessed the time effort combined with MAG output using VAMB, DASTool, and metaWRAP, respectively, as well as BASALT at each module. Overall, VAMB, DASTool, metaWRAP, and BASALT spent 4.6 h, 9.2 h, 29.7 h, and 41.3 h to finish the entire procedures, respectively, while BASALT spent a shorter time to finish up to Refinement Module (20.5 h) than metaWRAP (29.7 h) with better MAG yields (Supplementary Fig. 2A). In addition, BASALT MAGs mapped 64.9%, 72.2%, and 77.6% of raw sequencing reads when finishing Bin Selection, Refinement, and Gap Filling modules, respectively, whereas VAMB, DASTool, and metaWRAP generated MAGs mapped 61.8%, 49.6%, and 69.0% of raw sequencing reads, respectively (Supplementary Fig. 2B). These results suggested that BASALT had higher read usage efficiency than other tools.

To further evaluate BASALT efficiency in processing different types of assemblies, BASALT was compared with other tools in processing CAMI-high and CAMI-medium SRS datasets co-assembled using SPAdes and MEGAHIT. We found that binning efficiency was significantly higher with BASALT than other binning tools, resulting in a higher number of nonredundant, high-quality MAGs than other leading tools (Supplementary Figs. 3 and 4 and Supplementary Data 3). Overall, these findings suggested that BASALT can produce more MAGs with better quality than other tools, regardless the microbial diversity (low or moderate in this case) or input assembly types.

## BASALT genome retrieval from high-diversity environmental samples

To further test the quality of BASALT binning and genome extraction in high microbial diversity environmental samples, we analyzed four metagenomic samples obtained from Aiding Lake sediments. Nonpareil curves showed that the diversity of Aiding Lake sediment samples was higher than that in the CAMI-high dataset, and close to that in soil samples (Supplementary Fig. 5)[41], suggesting that the Aiding Lake sediment samples contained a highly complex microbial community. As Aiding Lake sediment samples only contained SRS, we further supplemented nine datasets, including four datasets with both SRS and LRS, and five PacBio High-Fidelity (HiFi) datasets, to assess the performance of BASALT on real samples. The four SRS + LRS datasets comprised a subset dataset from human gut microbiome (ten Illumina SRS samples, 204 GB in total, ten Oxford Nanopore (ONT) LRS samples, 113.6 GB in total)[42], a subset dataset from marine plankton microbiome (four Illumina SRS samples, 263.8 GB in total, and four Pacbio LRS samples, 91.6 GB in total)[43], a dataset from activated sludge microbiome (two Illumina SRS samples, 245.6 GB in total, and three ONT samples, 105.8 GB in total)[44], and a dataset from Antarctic soil microbiome (one Illumina sample 67.2 GB, one ONT sample 83.5 GB)[45]. The five PacBio HiFi datasets comprised a human gut microbiome (five samples, 182.6 GB in total)[46], a sheep gut microbiome (one sample, 92.1 GB)[47], a chicken gut microbiome (three samples, 366.8 GB in total)[48], a hot spring sediment microbiome (one sample 53.2 GB)[49], and an anaerobic digester microbiome (one sample 28.6 GB)[50]. Details of the above datasets were provided in Supplementary Data 9.

Using SRS assemblies, BASALT produced 557 nonredundant MAGs (completeness = 80.8% ± 12.57%, contamination = 1.45% ± 1.44%, quality = 73.2 ± 12.38) from Aiding Lake sediment samples, including 155 high-quality MAGs (completeness ≥90%, contamination ≤5%),

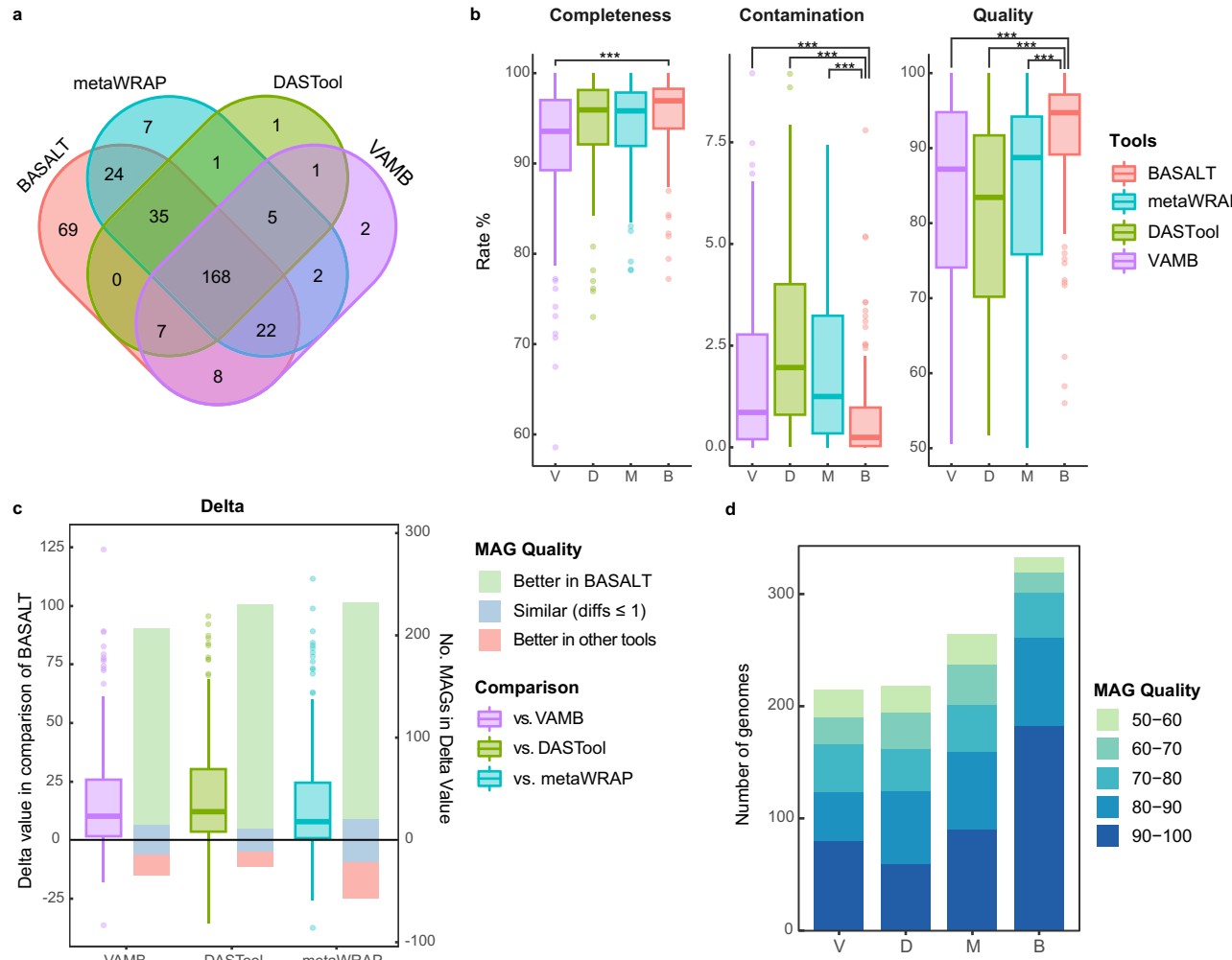

**Fig. 3 | Comparison of BASALT with other binning tools for processing the CAMI-high dataset. a** Venn diagram showing the number of MAGs with quality ≥50 (completeness−5* contamination ≥ 50) recovered using BASALT (red), metaWRAP (cyan), DASTool (green) or VAMB (purple). **b** Completeness, contamination, and quality scores of shared 168 MAGs recovered using BASALT (B, red), DASTool (D, green), metaWRAP (M, cyan), or VAMB (V, purple). MAGs recovered using BASALT had significantly higher completeness (compared to VAMB, Tukey test, Benjamini−Hochberg adjusted $P < 1 \times 10^{-7}$) and significantly lower contamination (compared to all other tools, Tukey test, Benjamini−Hochberg adjusted $P < 1 \times 10^{-7}$), resulting in significantly higher quality scores compared to VAMB, DASTool and metaWRAP (Tukey test, Benjamini−Hochberg adjusted $P < 1 \times 10^{-7}$). **c** Pairwise

comparison of MAGs shared between BASALT and VAMB (205 MAGs), BASALT and DASTool (210 MAGs), BASALT and metaWRAP (249 MAGs). BASALT obtained more MAGs with higher-quality scores than other tools. BASALT recovered more MAGs with higher quality (light green) than MAGs with lower (light red) or similar (difference ≤1, light blue) quality scores. **d** Number of MAGs recovered from CAMI-high dataset using DASTool (MaxBin2, CONCOCT, and MetaBAT2, MCM), VAMB, metaWRAP (MCM), or BASALT. Bar colors from light to dark indicate increasing MAG quality scores. The boxplot shows the distribution of data, the central dot in the box represents the median, the box bounds represent the 25th and 75th percentiles, and whiskers represent the minima to maxima values.

checked using CheckM[51]. Based on our above results showing meta-WRAP could generate more and higher-quality MAGs than VAMB and DASTool in CAMI-high data, we used metaWRAP for comparison with BASALT in genome extraction from Aiding Lake samples. We found that processing with metaWRAP yielded 392 nonredundant MAGs (completeness = 71% ± 13.2%, contamination = 2.4% ± 1.4%, quality = 58.9 ± 12.4), including 79 high-quality MAGs. Among the MAGs produced by BASALT and metaWRAP, 320 were determined to be identical based on ≥99% ANI and ≥60% alignment fraction (AF). In comparison between MAGs produced by BASALT and metaWRAP, BASALT obtained 96% more high-quality MAGs than metaWRAP (Supplementary Fig. 7 and Supplementary Data 10). Among shared MAGs, BASALT could also produce higher-quality MAGs than meta-WRAP (Supplementary Fig. 8A). Similar results were also obtained from the other nine real sample datasets (see details in Supplementary Notes). Among non-identical MAGs, BASALT produced 237 unique MAGs, while metaWRAP produced 72. Furthermore, analysis of

average coverage for each MAG revealed that BASALT could produce significantly more bins at lower coverage than metaWRAP (Kruskal−Wallis test, $P < 2.2 \times 10^{-16}$), with quality scores ranging from 50 to 90 for low coverage MAGs (average coverage <10) (Fig. 4a and Supplementary Data 4). These results suggested that BASALT could also recover low-abundance genomes from different types of samples, including highly complex communities.

## BASALT expands opening-reading frame annotations from MAGs

To evaluate MAGs obtained from Aiding Lake sediments using BASALT or metaWRAP, we next performed ORF prediction in all MAGs obtained by each toolkit (Fig. 4b and Supplementary Data 4). While 941,662 ORFs were identified in metaWRAP MAGs, a total of 1,466,017 ORFs were predicted in BASALT MAGs, a 47.6% increase compared to metaWRAP. In addition, the rate of unclassified ORFs (31.6%) was higher in BASALT than in metaWRAP ORFs (30.6%), suggesting that

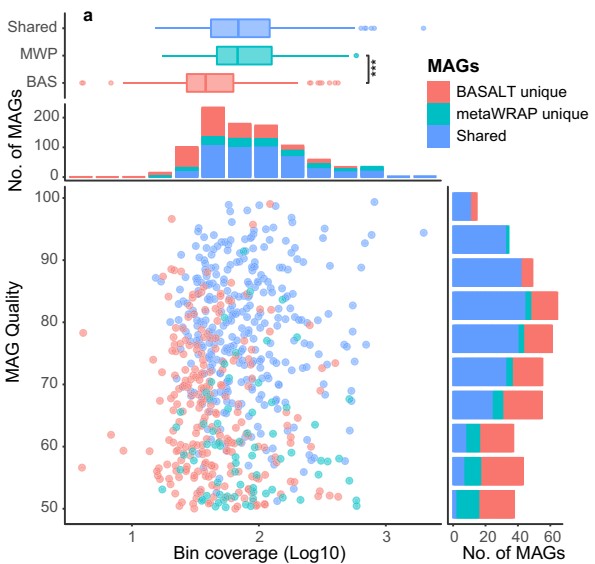

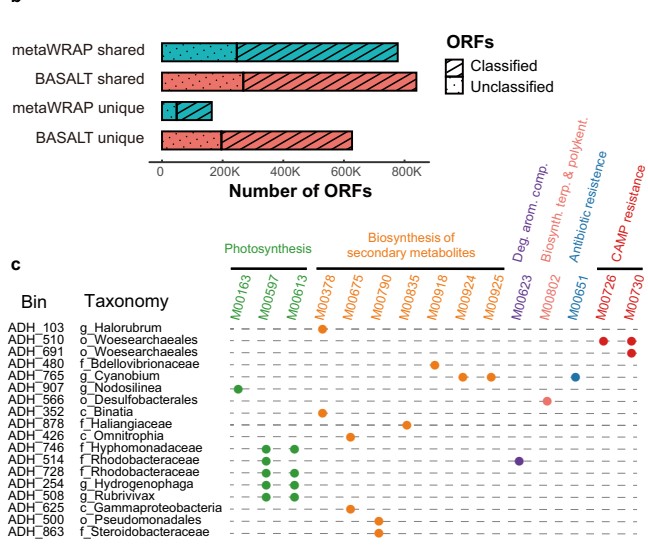

**Fig. 4 | Comparison of MAGs obtained by BASALT vs. metaWRAP from Aiding Lake sediment samples. a** Boxplot of bin coverage (top) in BASALT unique MAGs (red, $n = 237$), metaWRAP unique MAGs (cyan, $n = 72$), and shared MAGs (blue, $n = 320$). Benjamini−Hochberg adjusted $P$ values were calculated by the Dunn test. Significant differences between MAGs unique to BASALT and MAGs unique to metaWRAP were determined by $P = 7.49 \times 10^{-9}$. Breakdown of the number of MAGs in bin coverage (log10 transformed, second from top). Number of MAGs at each level of quality (lower right), and Scatter plot of MAGs quality and coverage (lower left). MAGs were unique to BASALT (red) or metaWRAP (cyan) with ANI ≥ 99% and AF ≥ 60% (blue). The average relative abundance of MAGs from BASALT was significantly lower than those from metaWRAP, and lower-coverage MAGs were evenly distributed across the range of MAG quality scores, indicating that BASALT could more effectively recover MAGs with lower coverage at any genome quality. The boxplot shows the distribution of data, the central dot in the box represents the median, the box bounds represent the 25th and 75th percentiles, and whiskers represent the minima to maxima values. **b** Summary of ORFs predicted in MAGs obtained by BASALT (red) or metaWRAP (cyan). **c** Modules only found in BASALT with corresponding MAGs. Green: photosynthesis; orange: biosynthesis of secondary metabolites; purple: degradation of aromatic compounds; pink: biosynthesis of terpenoids and polyketides; blue: antibiotic resistance; red: cationic antimicrobial peptide (CAMP) resistance.

BASALT could recognize slightly more putative functions in the Aiding Lake sediment metagenomic data. Among the shared MAGs, BASALT had 36,939 (4.7%) more ORFs than metaWRAP (Fig. 4b and Supplementary Data 4), further supporting the higher resolution of BASALT analysis. Similarly, more ORFs were also identified in BASALT MAGs on marine and human gut datasets compared to metaWRAP MAGs, regardless the MAGs were shared or unique (see details in Supplementary Notes).

In the context of metabolic pathways in Aiding Lake MAGs, 13 complete modules from 18 MAGs were unique to BASALT, including photosynthesis, biosynthesis of secondary metabolites, biosynthesis of terpenoids and polyketides, degradation of aromatic compounds, antibiotic resistance, and cationic antimicrobial peptide (CAMP) resistance (Fig. 4c and Supplementary Data 5). This finding suggested that BASALT had a higher capacity for acquiring complete modules from complex metagenomic data than metaWRAP.

## BASALT identifies class-level microbial lineages undetectable with other tools

To identify the taxonomy of MAGs obtained from Aiding Lake sediments using BASALT or metaWRAP, The 557 MAGs produced by BASALT and 392 MAGs by metaWRAP were annotated by searching against the Genome Taxonomy Database (GTDB)[39]. MAGs obtained from BASALT revealed the presence of 54 phyla (46 bacterial and 8 archaeal). The bacterial MAGs were mainly distributed in Patescibacteria, Acidobacteriota, Proteobacteria, Myxococcota, Desulfobacterota clade, PVC, FCB and Terrabacteria clades, while archaeal MAGs spanned the Halobacteriota, Asgardarchaeota, Thermoplasmatota, Thermoproteota, TACK and DPANN clades (Fig. 5 and Supplementary Data 6). However, 54.6% bacterial (258 MAGs) and 54.1% archaeal (46 MAGs) could not be assigned to any known genera in the GTDB, suggesting that Aiding Lake sediments might contain a

preponderance of uncharacterized genetic material. MAGs obtained from metaWRAP comprised 45 phyla (37 bacterial and 8 archaeal), which were all covered by BASALT MAGs. Notably, nine bacterial phyla were uniquely found in BASALT MAGs, including Deinococcota, Desulfobacterota_B, FEN-1099, Firmicutes_E, Fusobacteriota, Sumerlaeota, UBA3054, UBA6262, and UBP6. At lower taxonomic levels, a total of 21 bacterial classes and 2 archaeal orders were exclusively detected among BASALT MAGs (Fig. 5 and Supplementary Data 6), with the completeness of these MAGs ranged from 50.93 to 96.80% (mean = 73.42%), and the contamination ranged from 0 to 4.47% (mean = 1.29%) (Supplementary Data 8). Eleven of these 21 classes exclusive to BASALT were unclassified or phylogenomically close to a tentatively assigned class, suggesting that BASALT was capable of screening new lineages at high taxonomic levels from these sequencing data. Two MAGs from phyla Margulisbacteria and Cyanobacteria were obtained uniquely by BASALT in marine datasets (Supplementary Fig. 10), while lineages obtained uniquely by BASALT were underclass level in human gut datasets (Supplementary Fig. 11). In summary, BASALT was capable of acquiring more MAGs, as well as more MAGs from different taxonomic groups than metaWRAP, from the same dataset, suggesting the potential to substantially expand the scope of distinct branches in the tree of life.

In these data, we identified 174 MAGs that were closely related to the phylum Patescibacteria (also known as candidate phyla radiation/CPR superphylum) and the archaeal superphylum DPANN, among which 92.6% (101/109) of the Patescibacteria MAGs and 100% (65/65) of the DPANN MAGs could not be assigned or were assigned to a candidate family. Functional analysis at the ORF level revealed the presence of specific pathways that were rarely reported in previous studies. For example, we found two archaeal MAGs in phylum Nanoarchaeota that possessed putative CAMP resistance modules. In addition, two MAGs classified as genus *Prometheoarchaeum* (class Lokiarchaeia, phylum

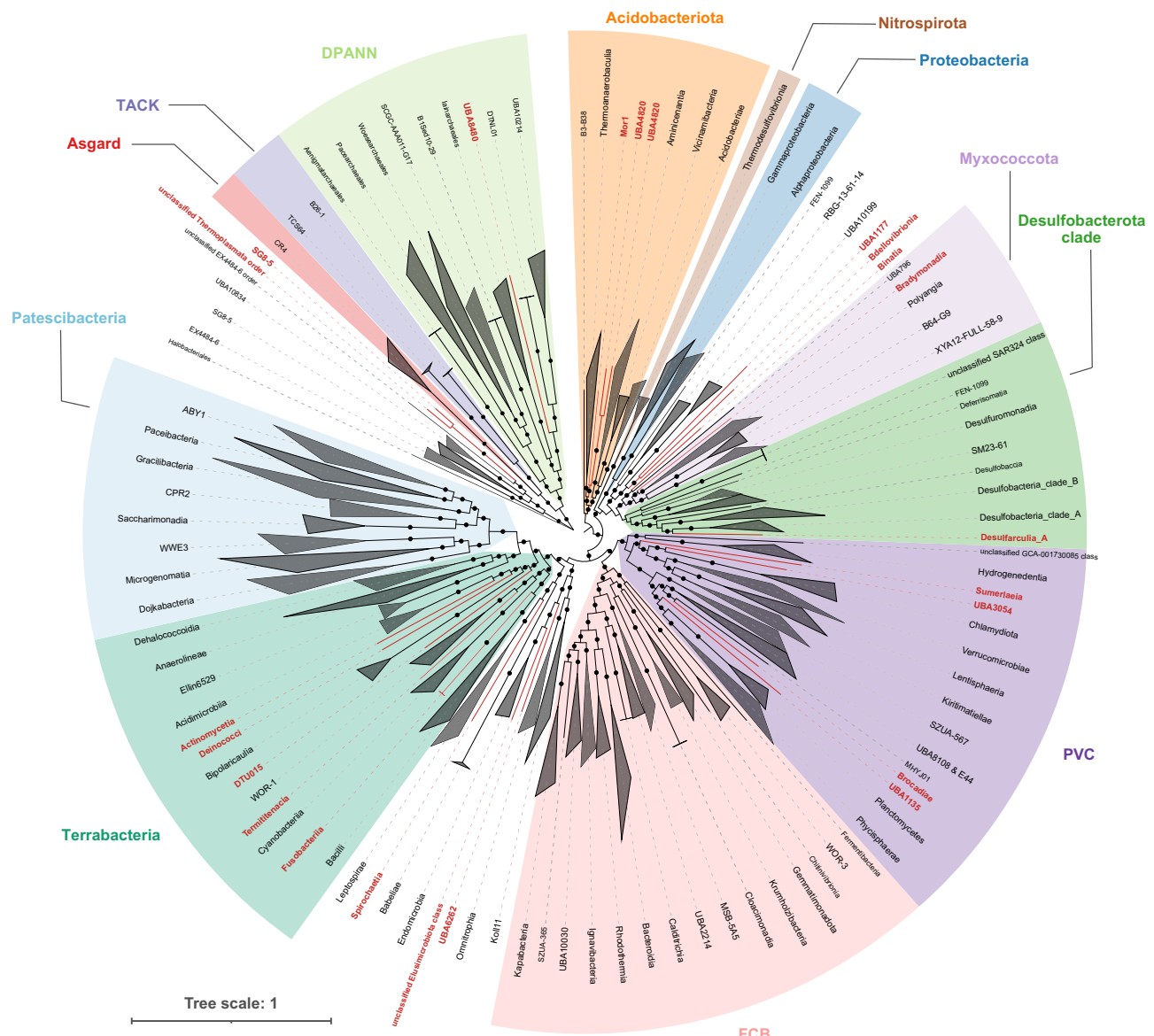

**Fig. 5 | Phylogenetic trees of bacterial and archaeal MAGs based on 120 and 122 concatenated marker genes, respectively.** Unrooted phylogenetic trees were constructed using IQ-TREE with 1000 bootstraps and best-fit models Q.pfam+R10 and LG + F + R8, respectively. Black dots in the middle of branches indicate >50% bootstrap support, and taxa highlighted in red indicate unique lineages at the class (bacteria) or order (archaea) levels obtained by BASALT.

Asgardarchaeota) were also observed in these samples, which have only been reported to date in marine samples. These analyses cumulatively provide a proof-of-concept demonstration of the high-resolution/high-quality/metagenomic sequence analysis with BASALT, and reveal the depth of complexity that has been likely overlooked with previous computational tools.

## Discussion

In this study, we demonstrated the use of BASALT for obtaining more and higher-quality genomes from low diversity or high-complexity whole-metagenome sequence data compared with other widely used toolkits. In particular, higher bin quality produced by BASALT enabled more bins obtained at MAG level than other tools, not only in low-abundant genomes but across all coverage levels, resulting in phylogenies contained more branches at all taxonomic levels (Figs. 4 and 5 and Supplementary Data 4). More importantly, the increase in binning efficiency suggests that information is extracted from the sequencing data, which will not only improve analyses of new sequencing data, but

will also facilitate more rigorous re-analyses of published datasets and publicly available binned data. BASALT also increases the usage efficiency of low-abundance reads by integrating multiple inputs of single assemblies and/or co-assemblies, enabling the recovery of high-quality genomes from low coverage, high-complexity community samples[4,32,52]. In addition to binning and post-binning refinement, BASALT contains additional functions that enable the processing and management of inputs from different stages of processing, including raw sequences, contigs, unpolished bins, and even polished bins, which enables dereplication and refinement of binsets generated by other tools (see details in Supplementary Notes). Overall, these results suggest that BASALT performs better than metaWRAP and similar tools in highly complex samples, which could, to some extent, reduce the burden of data processing reported in the EMP project[4]. Future development of BASALT could see the extension of input data types beyond the short-read or long-read sequencing data, to accommodate emerging technologies, such as DNA stable isotope probing (SIP)[53], Hi-

C[47], Pore-C[54], and single-cell assembled genomes (SAGs)[55], among others, for targeted analyses.

For this proof-of-concept study, the CAMI datasets were selected instead of the newer CAMI II challenge datasets because of the higher microbial diversity in CAMI-high data compared to CAMI and CAMI II datasets[56] (Supplementary Fig. 5). The lower diversity CAMI-medium dataset was used for comparison[31,57]. Furthermore, we processed metagenomic data from Aiding Lake sediments with BASALT to demonstrate its analytical power in high-complexity samples (Supplementary Fig. 5), comparable with soil samples[58], since no such high-diversity benchmarking datasets are currently available in public databases. Accuracy is essential to draw robust conclusions from metagenome data, and such assessment tools are available to assess the quality of processed metagenomic data[31,51,57,59]. However, benchmark datasets, e.g., gold standard assemblies simulated by CAMISIM[60], published specifically for this purpose could not be implemented in our study because it does not reflect actual assembly quality, only the potential quality under ideal conditions[27]. In this study, we used assemblers such as SPAdes, MEGAHIT, and Opera-MS, to produce contigs instead of gold standard (i.e., simulated) assemblies. In addition, we developed a custom script (detailed in "Methods") for assessing MAG quality. This script utilizes genome ANI against corresponding reference genomes from the CAMI project. Our approach deviates from using AMBER[61] as the assessment tool because of AMBER's limitation to the gold standard simulated assemblies, which are not representative of most metagenomic analyses derived from environmental samples. For the environmental samples, CheckM[51] has been used to evaluate MAG quality based on the presence of taxonomic marker genes in the absence of a reference genome. Furthermore, we have integrated CheckM2[62], a machine learning-enhanced quality assessment tool into BASALT. This implementation is particularly pertinent for the analysis of HiFi datasets, ensuring BASALT's outputs are comparable with that of the MAG-HiFi pipeline (https://github.com/PacificBiosciences/pb-metagenomics-tools). Despite the advent of CheckM2, CheckM is still considered as a major software predominantly used for quality assessment in most of the studies[4,7,30,55,63,64]. Consistent with this widespread usage, and to maintain comparability with the metaWRAP pipeline, CheckM was utilized for quality assessment of the remaining real sample datasets in our study.

In addition to validating the toolkit, BASALT was also used in this work to recover a large number of genus-level, unclassified MAGs from lake sediment samples, including some recently discovered lineages such as Patescibacteria and DPANN (Fig. 5). The reported genomes of Patescibacteria and DPANN are both relatively small, with characteristically incomplete metabolic functions typical of microbial symbionts[63,65]. Interestingly, we found two MAGs in phylum Nanoarchaeota with putative CAMP resistance modules. Although these two MAGs have not reached 90% of completeness (81.07% and 89.25%, respectively), their low contamination rates (0.47% and 0.93%, respectively) suggested that annotated genes in these two MAGs were possibly originated from their own genomes. Archaea are known to be insusceptible to a wide range of antimicrobials[66,67], and previous studies mainly found that some archaeal microorganisms could produce CAMP-like peptides[68,69], the presence of a CAMP resistance module might be redundant per se. However, it might represent a key factor enabling symbiosis with microalgae or other eukaryotes, which needs further exploration. Overall, BASALT analysis further expanded our understanding of the evolution and ecology of these microbes and their interaction partners as part of the so-called microbial dark matter in the tree of life[70–73].

Furthermore, this analysis uncovered two previously undocumented Lokiarchaeia genomes in Aiding Lake sample sediments, belonging to the recently designated Asgardarchaeota phylum, which is putatively linked to the origin of eukaryotes[74]. To date,

candidate Lokiarchaeia genomes have all been recovered from deep-sea hydrothermal vent and marine sediment ecosystems[74–79]. Although one recent study reported finding candidate Lokiarchaeia in brackish lakes connected to the Black Sea[80], to our knowledge, these genomes represent the first members of phylum Lokiarchaeia detected in deep inland, non-marine samples. Previous genomic analyses of Lokiarchaeia and other candidate Asgardarchaeota species[17,77,78,81,82] have suggested that candidate Lokiarchaeia are likely adaptive to different marine environments through various metabolic pathways, such as lignin or protein degradation. Based on the relative isolation of Aiding Lake, it would be interesting to explore how these candidate Lokiarchaeia species were introduced or evolved to persist in the saline conditions of this inland saline lake, in comparison with other reported candidate Lokiarchaeia MAGs. Future comparison with MAGs/isolates from other geographically distinct niches will help to resolve the evolutionary history and survival mechanisms of these organisms.

## Methods

### Overview of BASALT

BASALT is a binning and post-binning bioinformatics tool that recovers, compares, and optimizes assembled genomes across series of assemblies generated from short-read, long-read, or hybrid platforms to produce high-quality MAGs. Although BASALT can function with only a single metagenomic dataset, the overall bin quality including MAG quality can be improved using multiple datasets and assemblies as inputs[36,83]. A set of nine programs, designed in-house, work in concert to carry out functions including Auto-binning, Bin selection, Best-bins grouping, Core Sequence Identification, Outlier Removal, Sequence Retrieval, Polishing, Restrained Overlap–Layout–Consensus (rOLC), and Reassembly (Fig. 1). These functions are packaged into four modules: Automated Binning, Bin Selection, Refinement, and Gap Filling.

BASALT is a command line software compiled in Python 3.0 scripts, with each of the above modules containing one or more algorithms/programs. As an automated tool running with a single command line interface, checkpoints in each BASALT module allow users to stop and restart at any checkpoint as needed. In addition, each module can be executed individually, enabling users to customize the preferences as appropriate for their specific dataset(s). Further details regarding the code and tutorials are available at Github (https://github.com/EMBL-PKU/BASALT).

### Automated binning

By importing multiple metagenomic assemblies, BASALT calculates coverage by mapping raw sequence reads to the assembled contigs, then performs automated binning using multiple prominent binning tools, such as MetaBAT2, Maxbin2, and CONCOCT[24–26], etc., each set with multiple thresholds, to generate the initial binsets. Since binning with multiple binners and each with multiple thresholds may generate redundant bins[84], a Bin merging program is implemented to merge clustered contigs from potential redundant bins (identified by comprising same contigs) into a hybrid bin. Merged contigs in each hybrid bin are then dereplicated to generate hybrid binsets. A full list of available binning tools is provided in Supplementary Data 7.

### Bin selection

The Bin selection module is separated into two programs: Best-bins grouping and Core sequence identification (CSI). In the Best-bins grouping program, ANI is first calculated between each bin pair in the hybrid binsets. Bins at ANI ≥ 99% and AF ≥ 50% are grouped for further bin dereplication.

In the CSI algorithm, contig coverage is then calculated for each bin, after which a core coverage value ($x_i$) is calculated for each bin

using Eq. (1):

$$Q1 - k(IQR) < x_i < Q3 + k(IQR), \qquad (1)$$

in which Q1 and Q3 represent the 25th and 75th percentiles of all contigs, respectively, with IQR (interquartile ranges) calculated by Q3–Q1; $x_i$ represents the core coverage value range; and $k$ represents a constant for estimating IQR range, while BASALT performs core contigs identification with $k$ value at 0.

Core sequences in each bin that fall within the core coverage value range, $x_i$, are considered inliers, while sequences falling outside the $x_i$ range are outliers. To identify redundant bins, inliers for a given bin are pairwise aligned and compared within groups sorted by the Best-bins grouping program. In addition, a subset of inlier pairs that meet both the ANI ≥ 99% and length ≥1000 bps thresholds are further compared to determine the depth normalization ratio, $\bar{X}$, calculated by Eq. (2):

$$\bar{X} = \frac{1}{n} \sum_{i=1}^{n} \frac{cov_{p.inlierA1_i}}{cov_{p.inlierA2_i}}, \qquad (2)$$

In this formula, A1 and A2 represent the paired bins (e.g., Bin A1 and Bin A2); $n$ represent the number of inlier pairs that meet the ANI ≥ 99% and length ≥1000 bps thresholds; and $cov_{p.inlierA1_i}$ represent the coverage of a paired inlier sequence from Bin A1. While it is less likely that no inlier sequence pairs will be detected between two bins at ANI ≥ 99% and AF ≥ 50%, BASALT will retain both sequences as a nonredundant bin in this situation (Fig. 1).

Based on the depth normalization ratio, $\bar{X}$, normalized average inlier coverage of Bin A2 can be then calculated by multiplying $\bar{X}$ with the average inlier coverage of Bin A2, which is comparable with Bin A1. The delta coverage (Δ) between Bin A1 and Bin A2 is obtained by calculating the difference in average coverage of all inliers (including paired and unpaired inlier sequences) using Eq. (3):

$$\Delta = |\mu_{inlierA1} - \bar{X} \cdot \mu_{inlierA2}|, \qquad (3)$$

in which, $\mu_{inlierA1}$ represents the average coverage of all inlier sequences in Bin A1, while Δ < $w$ indicates redundant bins, with $w$ representing the threshold value trained by neural networks with multiple fully connected layers to distinguish absolute differences of average coverage (Δ). Details of the architecture of neural networks, loss function and model ensemble are available in Supplementary Methods. Finally, redundant bins are removed, while nonredundant bins are retained for further processing with the Refinement module.

## Refinement

The BASALT refinement module contains two programs: Outlier Removal (OR) and Sequence retrieval (Fig. 1), as well as a Polishing program for use with LRS data. The OR algorithm effectively removes contaminated sequences but avoids the simultaneous removal of multi-copy sequences from selected bins. Nonredundant bins are first assessed by two parameters to identify outlier contigs: tetranucleotide frequency (TNF) and coverage correlation coefficient (CCC). While TNF is widely used in microbial genomics studies[85], CCC can also be informative, calculated as the ratio of coverage values between input datasets using Eq. (4):

$$CCC_{ab} = cov_a / cov_b, \qquad (4)$$

In this equation, $cov_a$ and $cov_b$ are the coverages of a given contig in datasets $a$ and $b$, respectively. The CCC value of a contig is then calculated as a series of values for each pairwise comparison between datasets. Both core TNF and CCC are obtained following the core coverage method (Eq. (1)), while a multidimensional IQR is determined

using the $k$ value at 0, 0.5, 1, 2. Outlier sequences are then identified and removed to generate outlier-depleted bins (ODBs).

The un-binned sequences, including multi-copy genes that were potentially mis-assigned in the initial binning, are then retrieved by the Sequence Retrieval tool, which was designed to retrieve both un-binned short-read and long-read inlier contigs from the contigs pool, as previously assessed by OR. These inliers are then connected to ODBs via pair-end (PE) tracking[86]. Multiple iterations of contig connections are then performed to accommodate new inlier sequences in order to connect these sequences to bins refined in previous iterations. The iteration process is terminated when no new sequences can be connected to existing bins. To avoid generating potentially redundant bins in this step, CSI is conducted again to ensure that the refined binsets are nonredundant.

After sequence retrieval, the Polishing program is run to refine the mapped reads from polished bins. The polishing process is performed at this step to reduce computation time over that of polishing in the hybrid assembly process because only reads that mapped to the refined bins will be included in the analysis. Briefly, mapping is performed using raw SRS and LRS with refined binsets to extract mapped reads. Extracted sequences, especially LRS, are then polished using Pilon (v1.23)[87] through as many as ten iterations to correct sequence errors. To extract more mapped reads, a second round of mapping is then performed on polished bins using mapped reads from the polished bins along with unmapped reads. The polishing program is run for three cycles of polishing and mapping before running the gap-filling module.

## Gap filling

The BASALT Gap Filling module includes two restrained Overlap–Layout–Consensus (rOLC) processes with a reassembly step and a Polishing step in between, followed by a CSI step before the final bin-set is produced. The rOLC algorithm is designed to retrieve sequences not included in the BASALT binning and refinement processes, and fill sequence gaps using SRS/LRS data. First, the inlier sequences from redundant bins removed by CSI are reused after filtering with OR. Then, using a threshold of length ≥300 bp and ANI ≥99% for overlapping sequences, reads are overlapped with target bins in the Layout step, and merged with target bins in the Consensus step. To avoid filling contamination sequences into target bins, a restricted algorithm is applied in the Layout step if an overlapping sequence is (1) longer than target bin sequences at both ends, or (2) the length of a gap-filled sequence is <105% of the summed target bin sequence(s) length (Fig. 1). This strategy initially corrects long-read sequences by replacing low-quality base-pairs with N, and then re-aligns the corrected long reads from target bins with the original assembled sequences to retrieve more assembled sequences that were not previously included in the original bins.

After running the rOLC program, mapping is performed again to extract mapped reads from merged bins by Bowtie2[88], followed by a reassembly step using two state-of-art assemblers: SPAdes and IDBA[89,90] (note that hybridSPAdes[91] is used if LRS data are included). The Polishing program is run at this step to polish newly clustered reads generated in the rOLC and reassembly steps. Then, a second round of processing with rOLC is performed at a more stringent threshold overlap length of ≥500 bp and ANI ≥99% to further fill genomic gaps. A final round of CSI is conducted to select the best bin-set.

## Sample collection, DNA extraction and sequencing

Saline lake sediment samples were collected in July 2018 from Aiding Lake, located in an arid region in Turpan City, Xinjiang Uygur Autonomous Region (42°41′30″N 89°15′15″E). The map showing the location of Aiding Lake was generated using Google Map (Supplementary Fig. 1). Briefly, about 50 g of sediment samples ($n$ = 4) were randomly

collected at 0–10 cm depth in the lake and placed in sterile 50-ml conical centrifuge tubes. Samples were immediately placed on dry ice for transport to the laboratory and stored at −80 °C until further DNA extraction was performed.

Genomic DNA (gDNA) was extracted from the frozen sediment samples (~250 mg dry weight per sample) using a DNeasy PowerSoil Kit (Qiagen, Hilden, Germany), following the manufacturer's instructions. Extracted DNA was quality-checked by NanoDrop 2000 (Thermo Fisher Scientific, Waltham, MA, USA) and quantified by Qubit Fluorometer (Thermo Fisher Scientific). Quantity- and quality-checked gDNA samples were sent to Novegene Co., Ltd, Nanjing, China for shotgun metagenomics sequencing on the Illumina NovaSeq platform (2 × 150 bp paired end chemistry).

#### Sequence processing

To evaluate the efficiency of BASALT, standard Critical Assessment of Metagenome Interpretation (CAMI) datasets, including low-complexity (132 genomes, CAMI-medium) and moderate-complexity (596 genomes, CAMI-high) synthetic communities, were downloaded from (https://data.cami-challenge.org/participate)[31]. The rationale for selecting these benchmark datasets was based on the complexity of the synthetic communities compared to that of other CAMI datasets (Supplementary Fig. 5). Specifically, the experimental design used to generate the CAMI-high dataset included time-series sample which aligned with environmental research scopes, and thus we selected CAMI-high dataset for hybrid assembly to compare binning efficiency among different tools, whereas strictly SRS assembly was performed with the CAMI-medium dataset. Since LRS data was unavailable for the CAMI-high dataset, we simulated 50% of the CAMI-high sequences using CAMISIM[60] as ONT reads input to ensure a 2:1 SRS:LRS dataset size ratio (SRS: 150 GB, 2 × 150 paired-ended; LRS: 75 GB, No. contigs >1 kb: 1,945,842 ± 173.32, N50: 9,397.6 ± 2.42). To reflect the actual data analysis process, rather than using the gold assembly provided by CAMI, we used SPAdes (v3.14.1)[89] to individually assemble the simulated SRS reads into contigs in "–meta" mode, specifying k-mer sizes of 21, 33, 55, 77, and MEGAHIT (v1.2.9)[92], specifying k-mer sizes of 79, 99, 119, 149, which ultimately generated contigs >1000 bps. To improve binning performance in SRS-only analyses, reads were further co-assembled using SPAdes and MEGAHIT with the same assembly parameters as above. A hybrid assembly of the CAMI-high dataset was co-assembled by OPERA-MS[34] with/without a polishing step.

To compare with other binners/toolkits, both short-read-assembled and hybrid-assembled contigs were each processed with DASTool, VAMB, metaWRAP, and BASALT. To verify the accuracy of results obtained from the four binners/toolkits, the redundancy, completeness, and contamination of MAGs were calculated against that of standard CAMI genomes by aligning all sequences from a test bin to the gold standard genome using an in-house proprietary script, *Bin_quality_evaluation.py*, available on GitHub (https://github.com/EMBL-PKU/BASALT). In addition, this script also predicts ORFs and performs pairwise comparisons between the gold standard assembled genomes and test bins to detect contaminating ORFs. High-quality MAGs (completeness−5*contamination ≥50%) were retained for further statistical analysis, while bins that did not meet this quality cut-off were discarded.

BASALT and metaWRAP were individually used to process shotgun metagenomic sequences of Aiding Lake sediment samples, following the same procedure as that for CAMI-medium and CAMI-high datasets (SRS-only) above. MAG completeness and contamination were then estimated using CheckM version 1.1.3[51] with lineage-specific marker genes and default parameters, with only high- and medium-quality MAGs (completeness−5*contamination ≥50%) retained for further analyses.

#### Functional annotation

Opening Reading Frames (ORFs) of MAGs were predicted using Prodigal (v2.6.3) with default parameters[93], before annotation with Diamond (v2.0.11.149)[94], in "more-sensitive" mode, based on the Kyoto Encyclopedia of Genes and Genomes (KEGG) and NCBI-NR databases, at an *e* value cut-off of 1e-15.

#### Phylogenetic analysis

The GTDB-Tk v1.5.0[95] program was used to assign taxonomic classifications to the MAGs (release r202). Phylogenetic analyses were conducted for MAGs from both studies using IQ-TREE2[96] with best-fit model. Phylogenetic trees were visualized and edited in the iTOL v6 (https://itol.embl.de) online platform[97].

#### Statistical analysis and data visualization

Data organization and formatting were conducted using the R package "dplyr". Statistical analysis was conducted using the "vegan", "stats" and "FSA" R packages. Data visualization was performed using the R packages "ggplot2" and "venn".

#### Reporting summary

Further information on research design is available in the Nature Portfolio Reporting Summary linked to this article.

### Data availability

The CAMI benchmarking datasets are available at CAMI Challenge (https://data.cami-challenge.org/participate). The test genomes data generated in this study have been deposited in the CNCB-NGDC database (https://ngdc.cncb.ac.cn/) under accession code PRJCA014711. The metagenomic sequence data of Aiding Lake are deposited in the CNCB-NGDC database under accession code PRJCA014712. The accession codes of real datasets from other studies used in this study are downloaded from ENA sequence archive, including PRJEB52999 and PRJEB48021; NCBI sequence archive, including PRJNA820119, PRJNA648801, PRJNA681475, PRJNA750084, PRJNA595610, and PRJNA748109; DDBJ sequence archive DRR290133. Details of the sample accession codes are provided in Supplementary Data 9. The other data generated in this study are provided in the main text or the Supplementary Information.

### Code availability

All BASALT codes, including in-house scripts for quality checking against benchmarking datasets, are available at the GitHub repository (https://github.com/EMBL-PKU/BASALT) and Zenodo (https://doi.org/10.5281/zenodo.10653187).

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

## Acknowledgements

The authors would like to thank Dr. Chi Song, Dr. Chunfang Deng, Dr. Yang Wu, Dr. Baowei Chen, Ms. Xue Zhang, Ms. Yu Wang, and Mr. Tianyuan Zhang for technical assistance and valuable discussion. K.Y. acknowledges the National Key Research and Development Program of China (Project No. 2021YFA1301300), Nature Science Foundation of China (Project No. 51939009), and the Shenzhen Knowledge Innovation Program Basic Research Project (Project No. JCYJ20190808183205731 and JCYJ20220812103301001). L.Y. acknowledges the Nature Science Foundation of China (Project No. 62202014). B.L. received support from the Nature Science Foundation of China (Project No. 22176107). W.-Q.Z. thanks the Royal Society of New Zealand for the Marsden Fund (Project No. MFP-UOA2219).

## Author contributions

Z.Q. and K.Y. conceived the study. K.Y. built the scripts. Z.Q., C.L., L.Y., B.Lin, J.C., R.M., X.Q., L.Z. and K.Y. analyzed and interpreted the data. Z.Q., C.L., R.M., X.Q. and L.Z. performed metagenomic assembly and binning. L.Y., B.Lin and J.C. trained the neural networks. Z.Q., C.L. and K.Y. ran the benchmarks. C.L. released and maintained the BASALT software. Z.Q. drafted the manuscript. Z.Q., L.Y., Z.X., L.F., Y.Z., S.W., J.L., H.C., B.Li, C.Z., L.S., H.Q., Y.T., Z.Y., J.N., T.Z., J.Z., W.-Q.Z. and K.Y. revised the manuscript and provided critical discussions. W.-Q.Z., Y.Z., H.C., B.Li, L.Y., Y.L., B.C. and C.S. tested the software. All authors read and approved the final manuscript.

## Competing interests

The authors declare no competing interests.

## Additional information

[1]Eco-environment and Resource Efficiency Research Laboratory, School of Environment and Energy, Shenzhen Graduate School, Peking University, Shenzhen, China. [2]AI for Science (AI4S)-Preferred Program, Peking University, Shenzhen, China. [3]School of Electronic and Computer Engineering, Peking University, Shenzhen, China. [4]Peng Cheng Laboratory, Shenzhen, China. [5]Southern University of Sciences and Technology Yantian Hospital, Shenzhen, China. [6]Institute of Biomedicine and Biotechnology, Shenzhen Institute of Advanced Technology, Chinese Academy of Sciences, Shenzhen, Guangdong, China. [7]Department of Ocean Science and Engineering, Southern University of Science and Technology (SUSTech), Shenzhen, China. [8]Joint International Research Laboratory of Agriculture and Agri-Product Safety, the Ministry of Education of China, Yangzhou University, Yangzhou, China. [9]Environmental Microbiomics Research Center, School of Environmental Science and Engineering, Sun Yat-Sen University, Guangzhou, China. [10]School of Computer Science and Technology, Harbin Institute of Technology (Shenzhen), Shenzhen, Guangdong, China. [11]Department of Microbiology, University of Hong Kong, Hong Kong, China. [12]Shenzhen International Graduate School, Tsinghua University, Shenzhen, China. [13]Guangdong Provincial Key Laboratory of Marine Resources and Coastal Engineering, School of Marine Sciences, Sun Yat-sen University, Zhuhai, China. [14]Institute of Herbgenomics, Chengdu University of Traditional Chinese Medicine, Chengdu, China. [15]Wuhan Benagen Technology Co., Ltd, Wuhan, China. [16]Shenzhen Branch, Guangdong Laboratory of Lingnan Modern Agriculture, Genome Analysis Laboratory of the Ministry of Agriculture and Rural Affairs, Agricultural Genomics Institute at Shenzhen, Chinese Academy of Agricultural Sciences, Shenzhen, China. [17]State Key Laboratory of Chemical Oncogenomics, School of Chemical Biology and Biotechnology, Peking University Shenzhen Graduate School, Shenzhen, China. [18]College of Environmental Sciences and Engineering, Key Laboratory of Water and Sediment Sciences, Ministry of Education, Peking University, Beijing, China. [19]Department of Civil Engineering, University of Hong Kong, Hong Kong, China. [20]Institute for Environmental Genomics, University of Oklahoma, Norman, OK, USA. [21]Department of Civil and Environmental Engineering, Faculty of Engineering, University of Auckland, Auckland, New Zealand. [22]These authors contributed equally: Zhiguang Qiu, Ke Yu. ✉e-mail: yuke.sz@pku.edu.cn

