## [Peer Review File · Nature Communications]

BASALT refines binning from metagenomic data and increases resolution of genome-resolved metagenomic analysisReviewer #1 (Remarks to the Author):

Since I am a computational biologist and not a bioinformatician, I won't comment on software development aspects, but will provide a review from a user perspective.

The manuscript introduces BASALT (Binning Across a Series of Assemblies Toolkit), a tool that employs several binners with multiple thresholds to produce initial bins, then utilises neural networks to identify core sequences to remove redundant bins, followed by bin improvement. In brief, the tool consists of four main modules, i.e. binning, bin selection, bin refinement, and gap filling. The authors have demonstrated that their refinement and gap filling steps increase MAG completeness and reduce contamination. They also reported an increase in near complete bins.

Comparisons between BASALT and other published tools such as VAMB, DASTool and metaWRAP on the CAMI-high dataset, containing long-read and short-read sequencing data, are impressive, since i) BASALT recovered 50 MAGs not binned by other tools and ii) BASALT MAGs showed improved completeness, contamination and quality scores. I strongly suggest providing more information on the insert size/read length of these simulated "short and long-read" metagenomic data, created by CAMI. This is important to clarify since these CAMI data, generated over 5 years ago, likely do not resemble today's long-read sequencing data generated by ONT and/or PacBio HiFi long-read sequencing.

The comparison with metaWRAP using high complex communities from lake sediment samples is equally impressive. From Illumina short read (2×150 bp) assemblies, BASALT produced significantly more bins at lower coverage, and the conclusion that BASALT can recover low abundance genomes from highly complex communities is good news for everyone working with sediment and soil metagenomes. The fact that MAGs assigned to 9 bacterial phyla, 21 bacterial classes and 2 archaeal orders, were uniquely recovered by BASALT is remarkable. However, this result raises several questions about these MAGs. Are they all low abundances genomes, or are there any other reasons why only BASALT managed to recover these bins?

Also, Fig. 4a shows several MAGS with a very low average coverage. How confident are the authors about the quality of these bins?

From a user perspective, I am not convinced that low quality bins, e.g. with up to 20% contamination and less than 50% completeness, should be included in the analysis. These low-quality bins only provide beautified stats, e.g. larger numbers of recovered MAGs, but have little use otherwise, since low quality/high contamination MAGs are in general not used/ published by the scientific community. A more user-friendly approach will be to present only quality >50 MAGs in the main results, e.g. Fig. 3. The recovered low quality/high contamination MAGs can still be included in a suppl. figure or table if the authors wish to do so.

Minor comments:

Line 173 The comparison of the time required by each tool (BASALT, VAMB, DASTool, and metaWRAP) is a bit confusing. The overall time requirements for each tool should be provided and compared.

Line 199 High quality MAGs are defined as >90% compl, <5% contamination, according to MIMAG standards.

Line 249 Since both terms Patescibacteria and candidate phyla radiation (CPR) are used throughout the literature, I suggest saying something like "Patescibacteria (also known as the Candidate Phyla Radiation/CPR superphylum)"

Line 254 Regarding the "putative CAMP resistance modules", the authors might want to discuss their findings in light of the first CAMP-like peptide identified in Archaea and the potential of Archaea to produce antimicrobials: (<https://microbialcellfactories.biomedcentral.com/articles/10.1186/s12934->

015-0302-9)

Line 279 HiC and single-cell genomics are not new "sequencing" techniques, but rather alternative methods to recover microbial genomes from environmental samples.

Line 310 Asgardarchaeota are defined as a phylum, not a superphylum, in GTDB. Also note that the suffix -ota indicates a phylum level lineage. Subsequently, the Lokis are a class called "Lokiarchaeia" and should not be referred to as "Lokiarchaeota", which would indicate a phylum (-ota).

Line 318 "it is reasonable to speculate that these candidate Loki... species are not recently introduced, but rather have evolved to persist in the saline conditions of this inland saline lake."

This is an interesting idea. However, one could argue that the close phylogenetic relationship of the recovered Lokiarchaeia MAGs, i.e. they belong to the same genus (Prometheoarchaeum) as MAGs recovered from deep sea sediments, suggests that these archaea were in fact introduced comparably recently to the Aiding Lake sediments.

Line 347 "Bin merging program clusters bin contigs by identifying contig IDs, and these clustered contigs are then merged into hybrid bin-sets."

It is not clear to me how these hybrid bin-sets are generated and what they contain.

Line 388 I might have missed it, but is there any evaluation of mis-clustering of contigs into bin? Even despite the outlier removal (OR) and sequence retrieval steps, I assume that some contigs were miss-clustered in the CAMI dataset?

Fig. 5 What does "unclassified p__Elusimicrobiota" mean?

Reviewer #2 (Remarks to the Author):

The authors present BASALT, a binning pipeline that uses a combination of tools to obtain high quality MAGs from metagenomics. The authors compare BASALT to VAMB, DASTool, and metaWRAP and show that it returns better MAGs on the simulated CAMI dataset and compare to metaWRAP on an environmental dataset (lake sediment). They show improvements in all situations.

My impression is that this is a good tool, but the authors have not made their case as strongly as they could have. In particular, I think the evaluation on real-life data is too limited as I detail below.

#1. The authors mention long-read sequencing (LRS) as an advantage of their tool (e.g., Line 81), but the LRS data is only evaluated on the simulated dataset and not on real data.

This could be a strong selling point for BASALT, but with only simulated data, I do not think that the authors make their case. Furthermore, in the case of LRS data, the comparison points should be to binning tools/pipelines (see #3 below as well) that were designed for this data type.

This is actually not clear as the Methods only mention Illumina data (Line 452), but the main text mentions using LRS (Line 197).

#2. The evaluation on real data is based on 4 samples from the same environment. The authors could have exploited the fact that there are many publicly available metagenomes from all over the world and many different

environments.

#3. The authors compared BASALT to VAMB, DASTool, and metaWRAP. I think it's important to note that VAMB is a binning tool and metaWRAP (like BASALT) is a binning pipeline which incorporates binning tools as one of its components (it uses MetaBat2, MaxBin2, and CONCOCT, although the text on Line 345 is not clear and it seems that Table S7 is missing). In fact, it could incorporate VAMB itself.

#4. The analysis in the Section "BASALT identifies class-level microbial lineages undetectable with other tools" uses all the MAGs recovered without filtering for quality. Thus, the fact that BASALT identifies more lineages is not necessarily a good thing: it may be the case that BASALT is simply returning more low-quality MAGs.

Previously, the authors did report that the number of HQ MAGs is higher with BASALT, so this is unlikely to be the case, but it should be checked explicitly. It is otherwise possible that BASALT just recovers more HQ MAGs from the same lineages as metaWRAP (which could still be valuable, but the authors are making a stronger claim here).

In particular, the example that the authors use to illustrate the value of BASALT (Lines 253-254, "we found two archaeal MAGs in phylum Nanoarchaeota that possessed putative CAMP resistance modules") should be based on HQ MAGs only.

#5. I suggest the authors provide a bioconda package to install the software instead of the current approach.

Minor suggestion: I suggest rephrasing "screen out" on Line 210 as the English language is ambiguous and this could either mean "recover" or its opposite ("remove").

Response to reviewers' comments

On behalf of all authors, I'd like to thank the reviewers for their insightful and constructive comments. Following their suggestions, we have added two new real datasets with both short and long read sequencing data to support the performance of BASALT on real data analyses. In addition, following the advice and comments of reviewers, we have made several structural changes to the manuscript and provide a brief and general description of the changes here.

Below are the point-by-point responses to reviewers' comments:

Reviewer #1 (Remarks to the Author):

Since I am a computational biologist and not a bioinformatician, I won't comment on software development aspects, but will provide a review from a user perspective.

The manuscript introduces BASALT (Binning Across a Series of Assemblies Toolkit), a tool that employs several bidders with multiple thresholds to produce initial bins, then utilises neural networks to identify core sequences to remove redundant bins, followed by bin improvement. In brief, the tool consists of four main modules, i.e. binning, bin selection, bin refinement, and gap filling. The authors have demonstrated that their refinement and gap filling steps increase MAG completeness and reduce contamination. They also reported an increase in near complete bins.

Author response: Thank you, we appreciate and are grateful for the reviewer's positive comments.

1. Comparisons between BASALT and other published tools such as VAMB, DASTool and metaWRAP on the CAMI-high dataset, containing long-read and short-read sequencing data, are impressive, since i) BASALT recovered 50 MAGs not binned by other tools and ii) BASALT MAGs showed improved completeness, contamination and quality scores. I strongly suggest providing more information on the insert size/read length of these simulated "short and long-read" metagenomic data, created by CAMI. This is important to clarify since these CAMI data, generated over 5 years ago, likely do not resemble today's long-read sequencing data generated by ONT and/or PacBio HiFi long-read sequencing.

Author response: Thank you for your suggestion. The read length of simulated short reads of CAMI data is 150 bp paired-ended. The contig number and N50 of simulated long reads were summarized using QUAST.

We have added the relevant information of short- and long-read data in Lines 487 – 489:

"... we simulated 50% of the CAMI-high sequences using CAMISIM as ONT reads input to ensure a 2:1 SRS:LRS dataset size ratio (SRS: 150 GB, 2 × 150 paired-ended; LRS: 75 GB, No. contigs > 1 kb: 1,945,842 ± 173.32, N50: 9,397.6 ± 2.42)."

2. The comparison with metaWRAP using high complex communities from lake sediment

samples is equally impressive. From Illumina short read (2×150 bp) assemblies, BASALT produced significantly more bins at lower coverage, and the conclusion that BASALT can recover low abundance genomes from highly complex communities is good news for everyone working with sediment and soil metagenomes. The fact that MAGs assigned to 9 bacterial phyla, 21 bacterial classes and 2 archaeal orders, were uniquely recovered by BASALT is remarkable. However, this result raises several questions about these MAGs. Are they all low abundances genomes, or are there any other reasons why only BASALT managed to recover these bins?

Author response: Thanks for the positive comment and the questions.

Regarding the first question, “Are they all low abundance genomes?”, not all these MAGs uniquely recovered by BASALT were low-abundance genomes. As shown in Table S4, some of the low-abundance MAGs were assigned to phyla where both BASALT and metaWRAP can recover.

Regarding the second question, “Are there any other reasons why only BASALT managed to recover these bins?”, we believe that it is because BASALT’s unique algorithms and approaches to analyzing data. For instance, BASALT can recover bins with better completeness and less contamination. Following the MIMAG standards by Parks et al., only bins with quality (completeness – $5 \times$ contamination) above 50 can be considered MAGs. Therefore, even some bins were obtained by other software, such as metaWRAP, their algorithms might not be able to produce bins that can pass the quality threshold. Subsequently, low-quality bins were filtered out during the quality control/assessment step.

To make this point clearer, we added the following content in the discussion section in Lines 277 - 279:

“In particular, higher bin quality produced by BASALT enabled more bins obtained at MAG level than other tools, not only in low abundant genomes but across all coverage levels, resulting phylogenies contained more branches at all taxonomic levels (Figures 4 and 5, Table S4).”

3. Also, Fig. 4a shows several MAGS with a very low average coverage. How confident are the authors about the quality of these bins?

Author response: Thank you for raising this question. We have high confidence in these results because quality control steps, such as CheckM, were conducted to assess the accuracy of MAGs.

As demonstrated in other studies, even SPAdes or IDBA-UD could sometimes detect most SNPs and small indels in low sequencing depth (e.g., $\sim 7 \times$) datasets, resulting in a recovery of $> 90\%$ of genome fractions from the short-read assembly of a single genome^{1,2}. In the analysis of sediment metagenome datasets, BASALT was able to recover two MAGs with sequencing depth of $\sim 4 \times$ based on SPAdes assemblies, and a reassembly step using both SPAdes and IDBA-UD, finally reserving MAGs at completeness of 70% and 78%, respectively. Although the completeness of these two MAGs did not reach $> 90\%$, possibly due to the low sequencing depth ($\sim 4 \times$), we are confident that BASALT can recover low sequencing depth MAGs with an optimized assembly and binning strategy.

Additionally, in the analysis of sediment metagenome datasets, all bins were quality checked with CheckM, including these MAGs with very low coverage shown in Fig. 4a. Although MAG quality assessed by CheckM was based on the presence of taxonomic marker genes, which does not entirely reflect the real circumstances of microbial genomes in the environment, CheckM is still considered as a major software predominantly used for quality assessment in most of the studies.

To avoid confusion, we have changed the description “Average bin coverage (Log10)” in Fig. 4a to “Bin coverage (Log10)” indicating the total coverage of bins, and relevant content regarding CheckM usage was also discussed in the discussion section in Lines 311 – 314:

“Although MAG quality assessed by CheckM was based on the presence of taxonomic marker genes, which does not entirely reflect the real circumstances of microbial genomes in the environment, it is still considered as a major software predominantly used for quality assessment in most of the studies ^{4, 7, 30, 50, 57, 58}, and was therefore implemented to check MAG quality in Aiding Lake sediment data in the current study.”

4. From a user perspective, I am not convinced that low quality bins, e.g. with up to 20% contamination and less than 50% completeness, should be included in the analysis. These low-quality bins only provide beautified stats, e.g. larger numbers of recovered MAGs, but have little use otherwise, since low quality/high contamination MAGs are in general not used/published by the scientific community. A more user-friendly approach will be to present only quality >50 MAGs in the main results, e.g. Fig. 3. The recovered low quality/high contamination MAGs can still be included in a suppl. figure or table if the authors wish to do so.

Author response: Thank you for your suggestion. We have removed all bins with quality < 50 in Fig. 2. and Fig. 3, as well as updated the content in the Result section. Related data were correspondingly updated in Lines 158 – 165.

Minor comments:

5. Line 173 The comparison of the time required by each tool (BASALT, VAMB, DASTool, and metaWRAP) is a bit confusing. The overall time requirements for each tool should be provided and compared.

Author response: Thank you for your suggestion. We have rephrased the time requirement of each tool to avoid confusion in Lines 174 – 176:

“Overall, VAMB, DASTool, metaWRAP, and BASALT spent 4.6 h, 9.2 h, 29.7 h, and 41.3 h to finish the entire procedures, respectively, while BASALT spent a shorter time to finish up to Refinement Module (20.5 h) than metaWRAP (29.7 h) with better MAG yields (Figure S2A).”

6. Line 199 High quality MAGs are defined as >90% compl, <5% contamination, according to MIMAG standards.

Author response: Thank you for your suggestion. We have corrected the description of high-quality MAGs in Lines 201 – 218:

“BASALT produced 557 non-redundant MAGs (completeness = 80.8% ± 12.57%, contamination = 1.45% ± 1.44%, Quality = 73.2 ± 12.38), including 155 high-quality MAGs

(completeness \geq 90%, contamination \leq 5%), checked using CheckM ... We found that processing with metaWRAP yielded 392 non-redundant MAGs (completeness = $71\% \pm 13.2\%$, contamination = $2.4\% \pm 1.4\%$, Quality = 58.9 ± 12.4), including 79 high-quality MAGs.”

7. Line 249 Since both terms Patescibacteria and candidate phyla radiation (CPR) are used throughout the literature, I suggest saying something like “Patescibacteria (also known as the Candidate Phyla Radiation/CPR superphylum)”

Author response: Thank you. All “candidate phyla radiation (CPR)” have been changed to “Patescibacteria” (in Lines 262 – 264 and 318) as per the reviewer’s suggestion.

8. Line 254 Regarding the “putative CAMP resistance modules”, the authors might want to discuss their findings in light of the first CAMP-like peptide identified in Archaea and the potential of Archaea to produce antimicrobials:

(<https://microbialcellfactories.biomedcentral.com/articles/10.1186/s12934-015-0302-9>)

Author response: Thanks for your advice. We have added a discussion of the first CAMP-like peptide identified in Archaea and the potential of Archaea to produce antimicrobials in Lines 323 – 327:

“Archaea are known to be unsusceptible to a wide range of antimicrobials^{43, 44}, and previous studies mainly found that some archaeal microorganisms could produce CAMP-like peptides^{58, 59}, the presence of a CAMP resistance module might be redundant *per se*. However, it might represent a key factor enabling symbiosis with microalgae or other eukaryotes, which needs further exploration.”

9. Line 279 HiC and single-cell genomics are not new “sequencing” techniques, but rather alternative methods to recover microbial genomes from environmental samples.

Author response: Thanks for your suggestion. We have changed the description of these sequencing techniques as follows in Line 290:

“Moreover, input data are not limited to short-read or long-read sequencing data, and can accommodate alternative technologies such as DNA stable isotope probing (SIP), Hi-C, and single-cell assembled genomes (SAGs), among others, for targeted analyses.”

10. Line 310 Asgardarchaeota are defined as a phylum, not a superphylum, in GTDB. Also note that the suffix -ota indicates a phylum level lineage. Subsequently, the Lokis are a class called “Lokiarchaeia” and should not be referred to as “Lokiarchaeota”, which would indicate a phylum (-ota).

Author response: Thank you. We have corrected all “Lokiarchaeota” to “Lokiarchaeia” (Line 268 – 269, 330 – 341).

11. Line 318 “it is reasonable to speculate that these candidate Loki... species are not recently introduced, but rather have evolved to persist in the saline conditions of this inland saline lake.” This is an interesting idea. However, one could argue that the close phylogenetic relationship

of the recovered Lokiarchaeia MAGs, i.e. they belong to the same genus (Prometheoarchaeum) as MAGs recovered from deep sea sediments, suggests that these archaea were in fact introduced comparably recently to the Aiding Lake sediments.

Author response: Thanks for your suggestion. Indeed, the recovered Lokiarchaeia MAGs from Aiding Lake and marine sediments belong to the same genus, which could be recently introduced from one side to the other. Currently we don't have sufficient biological or geological evidence to support the above claim. Therefore, we changed the statement in Lines 339 – 341: “Based on the relative isolation of Aiding Lake, it would be interesting to explore how these candidate Lokiarchaeia species were introduced or evolved to persist in the saline conditions of this inland saline lake, in comparison with other reported candidate Lokiarchaeia MAGs.”

12. Line 347 “Bin merging program clusters bin contigs by identifying contig IDs, and these clustered contigs are then merged into hybrid bin-sets.”

It is not clear to me how these hybrid bin-sets are generated and what they contain.

Author response: Thank you for pointing out this unclear sentence. We have rephrased this sentence to avoid confusion (in Lines 367 – 370):

“Since binning with multiple bidders and each with multiple thresholds may generate redundant bins, a Bin merging program is implemented to merge clustered contigs from potential redundant bins (identified by comprising the same contigs) into a hybrid bin. Merged contigs in each hybrid bin are then dereplicated to generate hybrid bin-sets.”

13. Line 388 I might have missed it, but is there any evaluation of mis-clustering of contigs into bin? Even despite the outlier removal (OR) and sequence retrieval steps, I assume that some contigs were miss-clustered in the CAMI dataset?

Author response: Thanks for your question. In the assessment of CAMI dataset, mis-clustered contigs were identified by aligning to the gold-standard CAMI genomes (described in Line 474 – 478). In the Outlier removal and Sequence retrieval programs, we used several factors such as sequencing depth, TNFs, and Coverage Correlation Coefficient to establish data matrices to train the Ensemble models. Currently, this model enables BASALT to remove about 95% of miss-clustered contigs. However, a potential loss of corrected-clustered contigs may occasionally occur. Therefore, BASALT implemented CheckM to find the best bin between the original and refined bins. This approach is currently significantly effective across all test datasets. Although the presence of mis-clustered contigs after refinement is inevitable, we will keep updating the models using more upcoming benchmarking datasets in further software updates to minimize the mis-clustering of contigs.

14. Fig. 5 What does “unclassified p__Elusimicrobiota” mean?

Author response: Thank you. In Fig. 5, branches were collapsed at class (bacteria) or order (archaea) levels. The “unclassified p__...” indicates that MAG(s) on this branch cannot be assigned to a known class/order. Therefore, we used a higher level (phylum/class) to label the corresponding branch. To make this clearer, we have substituted “unclassified p__...” and

“unclassified c__...” with “unclassified Elusimicrobiota class” “unclassified ... order” ... to avoid confusion.

Reviewer #2 (Remarks to the Author):

The authors present BASALT, a binning pipeline that uses a combination of tools to obtain high quality MAGs from metagenomics. The authors compare BASALT to VAMB, DASTool, and metaWRAP and show that it returns better MAGs on the simulated CAMI dataset and compare to metaWRAP on an environmental dataset (lake sediment). They show improvements in all situations.

My impression is that this is a good tool, but the authors have not made their case as strongly as they could have. In particular, I think the evaluation on real-life data is too limited as I detail below.

Author response: Thank you. We are grateful for reviewer’s positive comments and valuable suggestions to strengthen our manuscript.

1. The authors mention long-read sequencing (LRS) as an advantage of their tool (e.g., Line 81), but the LRS data is only evaluated on the simulated dataset and not on real data. This could be a strong selling point for BASALT, but with only simulated data, I do not think that the authors make their case. Furthermore, in the case of LRS data, the comparison points should be to binning tools/pipelines (see #3 below as well) that were designed for this data type. This is actually not clear as the Methods only mention Illumina data (Line 452), but the main text mentions using LRS (Line 197).

Author response: Thanks for your suggestions. We have added two real datasets with both SRS and LRS data available (details described in response to #2) to support the use of the actual data section.

We are not aware of any publically available binning/binning refinement program that exploiting LRS or SRS+LRS for metagenome binning. Therefore, to make appropriate comparisons, we used the identical dataset(i.e., same assemblies sourced via SPAdes, MEGAHIT, or Opera-MS) for the comparison between BASALT and other tools to avoid biases that occurred at the assembly step.

We also thank you for pointing out the mistake in Line 197, as only SRS data was available in the lake sediment samples in this study. We have corrected the description (now in Line 201) as “Using SRS assemblies, BASALT produced ...”.

2. The evaluation on real data is based on 4 samples from the same environment. The authors could have exploited the fact that there are many publicly available metagenomes from all over the world and many different environments.

Author response: Thanks for your suggestion. We have added two real datasets with both SRS and LRS data, from human gut metagenome and marine plankton metagenome, respectively. Assembly and binning procedures were the same as for CAMI-high dataset.

Similar to the result of lake sediment analyses, we found BASALT generated more and better-quality MAGs. Additionally, unique lineages were also found in marine datasets at the phylum level.

These results strengthened the better performance of generating high-quality MAGs using BASALT. Detailed analyses were provided in the Results and Supplementary sections (Lines 195 – 200, 213 – 216, 226 – 228, 256 – 258, and Supplementary Materials)

3. The authors compared BASALT to VAMB, DASTool, and metaWRAP. I think it's important to note that VAMB is a binning tool and metaWRAP (like BASALT) is a binning pipeline which incorporates binning tools as one of its components (it uses MetaBat2, MaxBin2, and CONCOCT, although the text on Line 345 is not clear and it seems that Table S7 is missing). In fact, it could incorporate VAMB itself.

Author response: Thanks for your advice. We chose MetaBAT2, MaxBin2, and CONCOCT as binners for benchmarking comparison because some other popular pipeline tools (e.g., DASTool, metaWRAP) set these three binners as default (and only these three binners available in metaWRAP). BASALT has incorporated 6 binners in the latest version (binner list provided in Table S7), including VAMB, but only used MetaBAT2, MaxBin2, and CONCOCT for comparison in this study.

We have now clarified that VAMB is a binner, while the other two tools (DASTool and metaWRAP) are pipelines in Lines 157 – 158: “we used the CAMI benchmarking dataset to compare the outputs of BASALT with two binning pipelines (DASTool and metaWRAP) and a recently developed binner VAMB.”

4. The analysis in the Section "BASALT identifies class-level microbial lineages undetectable with other tools" uses all the MAGs recovered without filtering for quality. Thus, the fact that BASALT identifies more lineages is not necessarily a good thing: it may be the case that BASALT is simply returning more low-quality MAGs. Previously, the authors did report that the number of HQ MAGs is higher with BASALT, so this is unlikely to be the case, but it should be checked explicitly. It is otherwise possible that BASALT just recovers more HQ MAGs from the same lineages as metaWRAP (which could still be valuable, but the authors are making a stronger claim here).

In particular, the example that the authors use to illustrate the value of BASALT (Lines 253-254, "we found two archaeal MAGs in phylum Nanoarchaeota that possessed putative CAMP resistance modules") should be based on HQ MAGs only.

Author response: Thank you for the suggestions. We have high confidence in these results because quality control steps, such as CheckM, were conducted to assess the accuracy of MAGs.

We have summarized the status of MAGs from the lineages undetectable with other tools. The completeness of these MAGs ranged from 50.9% to 96.8% (mean = 73.4%), and the contamination ranged from 0% to 4.47% (mean = 1.29%), suggesting that these MAGs from the lineages uniquely found by BASALT were of good MAG quality. To clarify this point, we

have added detailed stats of completeness and contamination in the text (Line 251 – 253), as well as provided Table S8 for reference.

We also checked the completeness and contamination of two Nanoarchaeota MAGs detected with putative CAMP resistance modules. The completeness of these two MAGs were 81.07% and 89.25%, and the contamination were 0.47% and 0.93%, respectively. Although the completeness of these two MAGs did not reach 90%, low contamination rates (< 1%) suggested that annotated genes in these two MAGs were possibly originated from their own genomes.

To further clarify this in the manuscript, we added discussions in Lines 320 – 327:

“Interestingly, we found two MAGs in phylum Nanoarchaeota with putative CAMP resistance modules. Although these two MAGs have not reached 90% of completeness (81.07% and 89.25%, respectively), their low contamination rates (0.47% and 0.93%, respectively) suggested that annotated genes in these two MAGs were possibly originated from their own genomes. While Archaea are known to be insusceptible to a wide range of antimicrobials^{43, 44}, and previous studies mainly found that some archaeal microorganisms could produce CAMP-like peptides^{58, 59}, the presence of a CAMP resistance module might be redundant *per se*. However, it might represent a key factor enabling symbiosis with microalgae or other eukaryotes, which needs further exploration.”

5. I suggest the authors provide a bioconda package to install the software instead of the current approach.”

Author response: Thank you. We have simplified the BASALT installation procedure. Now the BASALT installation can be done with only two steps:

- 1) Download installation files: “git clone <https://github.com/EMBL-PKU/BASALT.git>”
- 2) Setting up the conda environment: “conda env create -n BASALT --file basalt_env.yml”

Detailed installation procedure can be found on Github: <https://github.com/EMBL-PKU/BASALT>.

6. Minor suggestion: I suggest rephrasing "screen out" on Line 210 as the English language is ambiguous and this could either mean "recover" or its opposite ("remove").

Author response: Thank you. We have changed “screen out” to “recover” (in Line 215).

1. Klein, J.D., Ossowski, S., Schneeberger, K., Weigel, D. & Huson, D.H. LOCAS—a low coverage assembly tool for resequencing projects. *PLoS One* **6**, e23455 (2011).
2. Forouzan, E., Maleki, M.S.M., Karkhane, A.A. & Yakhchali, B. Evaluation of nine popular de novo assemblers in microbial genome assembly. *J. Microbiol. Methods* **143**, 32-37 (2017).

Reviewer #1 (Remarks to the Author):

The authors have responded to all requests and have, e.g. added the relevant information of short- and long-read data, expended on the recovery of low sequencing depth MAGs, removed all bins with quality <50, updated the time requirements, corrected the description of high quality MAGs by referring to MiMAG standards, etc.

The only remaining question is why the authors used a custom script for determining MAG quality against the corresponding CAMI genomes rather than using the AMBER assessment tool? One should use AMBER, as the point of the CAMI datasets is that one knows which contigs are meant to go together. CheckM is appropriate, and indeed has become the standard, for metagenomic datasets.

Reviewer #2 (Remarks to the Author):

While I think this version is an improvement in many ways and the addition of the human and marine data do make the paper much stronger, I am still not fully convinced by the validation on real data. I would wish for more than a handful of samples (there are 100,000s of short read metagenomes out there and while the number of long-read ones is much smaller, it has really grown in the last few years) and I think they should be compared to a baseline that is designed for long-reads (when applicable).

The last sentence of the results is perhaps telling. The authors claim that their "analyses cumulatively provide a proof-of-concept demonstration of the high resolution/high quality/metagenomic sequence analysis with BASALT." I completely agree with them, but I would wish for more than a proof-of-concept demonstration (particularly for a journal such as Nature Communications).

ON THE CHOICE OF BASELINE

In my previous comments, I had pointed out that the authors had not compared against tools designed for long reads. The authors responded that they "are not aware of any publically available binning/binning refinement program that exploiting LRS or SRS+LRS for metagenome binning." Indeed, this field is very recent, but there are several tools already. For PacBio data, the company itself publishes a pipeline for metagenomics, which is very complete and could be seen as an alternative to BASALT:

<https://github.com/PacificBiosciences/pb-metagenomics-tools/tree/master/HiFi-MAG-Pipeline>

Note that although the pipeline is developed by a commercial company, it is (to the best of my knowledge), available as open-source and their benchmarking (much more extensive than the one in this manuscript) can be reproduced or used as a baseline. It is using DAS Tool, but using SemiBin2 (which has an LRS mode) as one of its binners.

MINOR COMMENTS

There is a lack of detail on how the human gut and marine samples were handled and even on their characteristics. A reader must go to Table S9 to even find out the number of samples used as only the total number of basepairs is shown in the main text. Initially I even thought that a single sample was present from each biome as the sequencing is relatively shallow: to sample a wider variety of sampling contexts and provide the readers with a better understanding of when BASALT can provide better results than the alternatives and its limitations.

Lines 222-4: "In addition, the rate of unclassified ORFs (31.6%) was higher in BASALT than in metaWRAP ORFs (30.6%), implying that BASALT could potentially recognize more putative functions

within the environmental metagenomic data." This reads too much into a single percentage point difference in a single study.

Line 253: "MAGs from the lineages uniquely found by BASALT were of good MAG quality" — In the previous sentence, the authors say that the completeness ranges from 50-97%, which includes many which are conventionally classified as "Medium quality" and not "good".

Line 291: I think the placement of the sentence starting with "Overall, these results suggest that BASALT performs better than metaWRAP" is misleading. It immediately follows a mention that BASALT can use other technologies to complement sequencing. However, there are no results in the manuscript about any of them (the one with which I have direct experience, Hi-C, is surprisingly tricky to take advantage of, for example).

Line 311: I do not understand the concern behind the comment "MAG quality assessed by CheckM was based on the presence of taxonomic marker genes, which does not entirely reflect the real circumstances of microbial genomes in the environment". I will also note that CheckM2 recently came out and it uses a completely different approach. I will note, though, that one of its limitations is that CheckM's estimates of contamination are less accurate for less complete genomes, which is relevant for the comment that the authors make on Line 321.

Response to Reviewers' Comments

We sincerely thank the reviewers for their dedicated time and insightful feedback. Their constructive comments have been invaluable in enhancing the quality of our work. In response to their recommendations, we have enriched our study by conducting benchmark analyses and incorporating seven additional datasets. These datasets include both short and long read sequencing data, notably featuring PacBio HiFi sequencing data, to robustly demonstrate the efficacy of BASALT in real data analyses. Furthermore, we have implemented several structural modifications to the manuscript in line with the reviewers' insightful advice. Below, we provide a concise overview of these significant changes. Below are our point-by-point responses to reviewers' comments:

Reviewer #1 (Remarks to the Author):

The authors have responded to all requests and have, e.g., added the relevant information of short- and long-read data, expended on the recovery of low sequencing depth MAGs, removed all bins with quality <50, updated the time requirements, corrected the description of high quality MAGs by referring to MiMAG standards, etc.

Author response: We sincerely thank the reviewer for the positive comments.

The only remaining question is why the authors used a custom script for determining MAG quality against the corresponding CAMI genomes rather than using the AMBER assessment tool? One should use AMBER, as the point of the CAMI datasets is that one knows which contigs are meant to go together. CheckM is appropriate, and indeed has become the standard, for metagenomic datasets.

Author response: Although CAMI datasets recommended AMBER for benchmarking comparison, we rationalized that it required gold standard assemblies as input for binning. Gold standard assemblies are simulated contigs produced at ideal circumstances and they do not reflect actual assembly quality using widely used assemblers such as SPAdes, MEGAHIT, and Opera-MS. This observation was also mentioned by Nissen *et al.* in VAMB¹. Hence, we used these widely used assemblers, such as SPAdes, MEGAHIT, and Opera-MS, to produce contigs instead of gold standard (*i.e.*, simulated) assemblies. CheckM assesses genome quality based on marker genes, which is robust without reference genomes. However, when standard reference genomes are given in simulated mock communities, such as CAMI datasets, a better approach is to evaluate bins against reference genomes, which compares not only marker genes but the entire sequences of genomes. Therefore, we developed the custom script to assess the entire sequences of bins based on ANI against the corresponding reference genomes. To make this statement clearer, we have modified the discussion in lines 319 – 331:

“In addition, we developed a custom script (detailed in the Methods section) for assessing MAG quality. This script utilizes genome ANI against corresponding reference genomes from the CAMI project. Our approach deviates from using AMBER

² as the assessment tool because of AMBER's limitation to the gold standard simulated assemblies, which are not representative of most metagenomic analyses derived from environmental samples. For the environmental samples, CheckM ³ has been used to evaluate MAG quality based on the presence of taxonomic marker genes in the absence of a reference genome. Furthermore, we have integrated CheckM2 ⁴, a machine learning enhanced quality assessment tool into BASALT. This implementation is particularly pertinent for the analysis of HiFi datasets, ensuring BASALT's outputs are comparable with that of the MAG-HiFi pipeline. Despite the advent of CheckM2, CheckM is still considered as a major software predominantly used for quality assessment in most of the studies ⁵⁻¹⁰. Consistent with this widespread usage, and to maintain comparability with the metaWRAP pipeline, CheckM was utilized for quality assessment of the remaining real sample datasets in our study."

Reviewer #2 (Remarks to the Author):

While I think this version is an improvement in many ways and the addition of the human and marine data do make the paper much stronger, I am still not fully convinced by the validation on real data. I would wish for more than a handful of samples (there are 100,000s of short read metagenomes out there and while the number of long-read ones is much smaller, it has really grown in the last few years) and I think they should be compared to a baseline that is designed for long-reads (when applicable).

The last sentence of the results is perhaps telling. The authors claim that their "analyses cumulatively provide a proof-of-concept demonstration of the high resolution/high quality/metagenomic sequence analysis with BASALT." I completely agree with them, but I would wish for more than a proof-of-concept demonstration (particularly for a journal such as Nature Communications).

Author response: We sincerely thank the reviewer for those comments and suggestions. In this revision, we added seven groups of datasets to further evaluate the performance of BASALT. These datasets included two SRS+LRS datasets (activated sludge and Antarctic soil) and five PacBio HiFi datasets (human gut, chicken gut, sheep gut, hot spring, and anerobic digester) designed for long-reads. Details of the comparison are provided below and in the revised manuscript.

ON THE CHOICE OF BASELINE

In my previous comments, I had pointed out that the authors had not compared against tools designed for long reads. The authors responded that they "are not aware of any publically available binning/binning refinement program that exploiting LRS or SRS+LRS for metagenome binning." Indeed, this field is very recent, but there are several tools already. For PacBio data, the company itself publishes a pipeline for metagenomics, which is very complete and could be seen as an alternative to BASALT:

<https://github.com/PacificBiosciences/pb-metagenomics-tools/tree/master/HiFi-MAG-Pipeline>

Note that although the pipeline is developed by a commercial company, it is (to the best of my knowledge), available as open-source and their benchmarking (much more extensive than the one in this manuscript) can be reproduced or used as a baseline. It is using DAS Tool, but using SemiBin2 (which has an LRS mode) as one of its binners.

Author response: We sincerely thank the reviewer for recommending a benchmark analysis with more datasets and against the HiFi-MAG pipeline. In accordance with this suggestion, we have expanded our analysis to include seven additional datasets, including two datasets with both SRS and LRS data and five datasets with PacBio HiFi data.

To facilitate a comprehensive comparison with the HiFi-MAG pipeline, we have integrated both SemiBin2 and CheckM2 into our BASALT framework. This integration enabled us to analyze the five HiFi datasets and directly compare our results with those produced by the HiFi-MAG pipeline. Consistent with our previous findings, BASALT demonstrated its capability to generate more high-quality MAGs compared to other available tools, including the HiFi-MAG pipeline, across all datasets.

We have included detailed analyses of these findings in both the Results section and the Supplementary Materials of our manuscript (Lines 199 – 210, 219 – 223, and Supplementary Materials).

MINOR COMMENTS

There is a lack of detail on how the human gut and marine samples were handled and even on their characteristics. A reader must go to Table S9 to even find out the number of samples used as only the total number of basepairs is shown in the main text. Initially I even thought that a single sample was present from each biome as the sequencing is relatively shallow: to sample a wider variety of sampling contexts and provide the readers with a better understanding of when BASALT can provide better results than the alternatives and its limitations.

Author response: Thanks for pointing this out. We have added details of real datasets in Lines 196 – 207:

“... we further supplemented nine datasets, including four datasets with both SRS and LRS, and five PacBio High-Fidelity (HiFi) datasets, to assess the performance of BASALT on real samples. The four SRS+LRS datasets comprised a subset dataset from human gut microbiome (ten Illumina SRS samples, 204 GB in total, and ten Oxford Nanopore (ONT) LRS samples, 113.6 GB in total) ¹¹, a subset dataset from marine plankton microbiome (four Illumina SRS samples, total 263.8 GB, four Pacbio LRS samples, 91.6 GB in total) ¹², a dataset from activated sludge microbiome (two Illumina SRS samples, 245.6 GB in total, and three ONT samples, 105.8 GB in total) ¹³, and a dataset from Antarctic soil microbiome (one Illumina sample 67.2 GB, and one ONT sample 83.5 GB) ¹⁴. The five PacBio HiFi datasets comprised a human gut microbiome (five samples, 182.6 GB in total) ¹⁵, a sheep gut microbiome (one sample 92.1 GB) ¹⁶, a chicken gut microbiome (three samples, 366.8 GB in total) ¹⁷, a hot spring sediment microbiome (one sample 53.2 GB) ¹⁸, and an anaerobic digester

microbiome (one sample 28.6 GB)¹⁹. Details of the above datasets were provided in Table S9.”

Lines 222-4: "In addition, the rate of unclassified ORFs (31.6%) was higher in BASALT than in metaWRAP ORFs (30.6%), implying that BASALT could potentially recognize more putative functions within the environmental metagenomic data." This reads too much into a single percentage point difference in a single study.

Author response: Thank you. We have corrected the sentence to avoid confusion in lines 236 – 237:

“In addition, the rate of unclassified ORFs (31.6%) was higher in BASALT than in metaWRAP ORFs (30.6%), suggesting that BASALT could recognize more putative functions in the Aiding Lake sediment metagenomic data.”

Line 253: "MAGs from the lineages uniquely found by BASALT were of good MAG quality" — In the previous sentence, the authors say that the completeness ranges from 50-97%, which includes many which are conventionally classified as "Medium quality" and not "good".

Author response: In accordance with this suggestion, we have removed the statement of “good MAG quality” and rephrased sentences in lines 262 – 265 as below:

“At lower taxonomic levels, a total of 21 bacterial classes and two archaeal orders were exclusively detected among BASALT MAGs (Figure 5, Table S6), with the completeness of these MAGs ranged from 50.93% to 96.80% (mean = 73.42%), and the contamination ranged from 0% to 4.47% (mean = 1.29%) (Table S8).”

Line 291: I think the placement of the sentence starting with "Overall, these results suggest that BASALT performs better than metaWRAP" is misleading. It immediately follows a mention that BASALT can use other technologies to complement sequencing. However, there are no results in the manuscript about any of them (the one with which I have direct experience, Hi-C, is surprisingly tricky to take advantage of, for example).

Author response: Thanks for the suggestion. To avoid the ambiguity, we have revised the sentences in lines 301 – 306:

“Overall, these results suggest that BASALT performs better than metaWRAP and similar tools in highly complex samples, which could, to some extent, reduce the burden of data processing reported in the EMP project⁵. Future development of BASALT could see the extension of input data types beyond the short-read or long-read sequencing data, to accommodate emerging technologies, such as DNA stable isotope probing (SIP)²⁰, Hi-C¹⁶, Pore-C²¹, and single-cell assembled genomes (SAGs)⁸, among others, for targeted analyses.”

Line 311: I do not understand the concern behind the comment "MAG quality assessed by CheckM was based on the presence of taxonomic marker genes, which does not entirely reflect the real circumstances of microbial genomes in the environment". I will

also note that CheckM2 recently came out and it uses a completely different approach. I will note, though, that one of its limitations is that CheckM's estimates of contamination are less accurate for less complete genomes, which is relevant for the comment that the authors make on Line 321.

Author response: Thanks for pointing this out. This sentence was meant to further explain why a custom script was used to determine MAG quality in the CAMI datasets instead of CheckM in the presence of reference genomes. While we agree that CheckM is appropriate and widely used in genome quality assessment without reference genomes, current sentences seem to cause confusion. We have modified the corresponding content in lines 323 – 331:

“For the environmental samples, CheckM³ has been used to evaluate MAG quality based on the presence of taxonomic marker genes in the absence of a reference genome. Furthermore, we have integrated CheckM2⁴, a machine learning enhanced quality assessment tool into BASALT. This implementation is particularly pertinent for the analysis of HiFi datasets, ensuring BASALT's outputs are comparable with that of the MAG-HiFi pipeline. Despite the advent of CheckM2, CheckM is still considered as a major software predominantly used for quality assessment in most of the studies⁵⁻¹⁰. Consistent with this widespread usage, and to maintain comparability with the metaWRAP pipeline, CheckM was utilized for quality assessment of the remaining real sample datasets in our study.”

References

1. Nissen, J.N. et al. Improved metagenome binning and assembly using deep variational autoencoders. *Nat. Biotechnol.*, 1-6 (2021).
2. Meyer, F. et al. AMBER: assessment of metagenome BinnERs. *Gigascience* **7**, giy069 (2018).
3. Parks, D.H., Imelfort, M., Skennerton, C.T., Hugenholtz, P. & Tyson, G.W. CheckM: assessing the quality of microbial genomes recovered from isolates, single cells, and metagenomes. *Genome Res.* **25**, 1043-1055 (2015).
4. Chklovski, A., Parks, D.H., Woodcroft, B.J. & Tyson, G.W. CheckM2: a rapid, scalable and accurate tool for assessing microbial genome quality using machine learning. *Nat. Methods* **20**, 1203-1212 (2023).
5. Nayfach, S. et al. A genomic catalog of Earth's microbiomes. *Nat. Biotechnol.*, 1-11 (2020).
6. Parks, D.H. et al. Recovery of nearly 8,000 metagenome-assembled genomes substantially expands the tree of life. *Nature microbiology* **2**, 1533-1542 (2017).
7. He, C. et al. Genome-resolved metagenomics reveals site-specific diversity of epibiotic CPR bacteria and DPANN archaea in groundwater ecosystems. *Nat Microbiol* **6**, 354-365 (2021).
8. Arikawa, K. et al. Recovery of strain-resolved genomes from human microbiome through an integration framework of single-cell genomics and metagenomics. *Microbiome* **9**, 1-16 (2021).
9. Chibani, C.M. et al. A catalogue of 1,167 genomes from the human gut archaeome. *Nature Microbiology* **7**, 48-61 (2022).
10. Pasolli, E. et al. Extensive unexplored human microbiome diversity revealed by over 150,000 genomes from metagenomes spanning age, geography, and lifestyle. *Cell* **176**, 649-662. e620 (2019).

11. Chen, L. et al. Short-and long-read metagenomics expand individualized structural variations in gut microbiomes. *Nature Communications* **13**, 3175 (2022).
12. Orellana, L.H., Krüger, K., Sidhu, C. & Amann, R. Comparing genomes recovered from time-series metagenomes using long-and short-read sequencing technologies. *Microbiome* **11**, 105 (2023).
13. Liu, L. et al. Charting the complexity of the activated sludge microbiome through a hybrid sequencing strategy. *Microbiome* **9**, 1-15 (2021).
14. Waschulin, V. et al. Biosynthetic potential of uncultured Antarctic soil bacteria revealed through long-read metagenomic sequencing. *The ISME journal* **16**, 101-111 (2022).
15. Kim, C.Y., Ma, J. & Lee, I. HiFi metagenomic sequencing enables assembly of accurate and complete genomes from human gut microbiota. *Nature communications* **13**, 6367 (2022).
16. Bickhart, D.M. et al. Generating lineage-resolved, complete metagenome-assembled genomes from complex microbial communities. *Nat. Biotechnol.* (2022).
17. Zhang, Y. et al. Improved microbial genomes and gene catalog of the chicken gut from metagenomic sequencing of high-fidelity long reads. *GigaScience* **11**, giac116 (2022).
18. Kato, S., Masuda, S., Shibata, A., Shirasu, K. & Ohkuma, M. Insights into ecological roles of uncultivated bacteria in Katase hot spring sediment from long-read metagenomics. *Frontiers in Microbiology* **13**, 1045931 (2022).
19. Sereika, M. et al. Oxford Nanopore R10. 4 long-read sequencing enables the generation of near-finished bacterial genomes from pure cultures and metagenomes without short-read or reference polishing. *Nat. Methods* **19**, 823-826 (2022).
20. Wasmund, K. et al. Genomic insights into diverse bacterial taxa that degrade extracellular DNA in marine sediments. *Nature Microbiology* **6**, 885-898 (2021).
21. Deshpande, A.S. et al. Identifying synergistic high-order 3D chromatin conformations from genome-scale nanopore concatemer sequencing. *Nat. Biotechnol.* **40**, 1488-1499 (2022).

Reviewer #1 (Remarks to the Author):

The authors have addressed all concerns and now explain why they deviated from the standard approach of using AMBER to assess MAGs.

Reviewer #2 (Remarks to the Author):

The authors have addressed my substantive concerns. I am now convinced that BASALT has value on real data.

I think the sentence "In addition, the rate of unclassified ORFs (31.6%) was higher in BASALT than in metaWRAP ORFs (30.6%), suggesting that BASALT could recognize more putative functions in the Aiding Lake sediment metagenomic data" is still reading too much into a small difference, but this is not a major issue.

Response to Reviewers' Comments

We sincerely thank the reviewers for their time and feedback. In response to the comments, we have slightly changed the sentence mentioned by Reviewer #2 to avoid overstatement. Below are our point-by-point responses to reviewers' comments:

Reviewer #1 (Remarks to the Author):

The authors have addressed all concerns and now explain why they deviated from the standard approach of using AMBER to assess MAGs.

Author response: We sincerely thank the reviewer for the positive comments.

Reviewer #2 (Remarks to the Author):

The authors have addressed my substantive concerns. I am now convinced that BASALT has value on real data.

Author response: We sincerely thank the reviewer for the constructive comments and suggestions.

I think the sentence "In addition, the rate of unclassified ORFs (31.6%) was higher in BASALT than in metaWRAP ORFs (30.6%), suggesting that BASALT could recognize more putative functions in the Aiding Lake sediment metagenomic data" is still reading too much into a small difference, but this is not a major issue.

Author response: Thank you. We have modified this sentence to "In addition, the rate of unclassified ORFs (31.6%) was higher in BASALT than in metaWRAP ORFs (30.6%), suggesting that BASALT could recognize slightly more putative functions in the Aiding Lake sediment metagenomic data" in line 273 to avoid the overstatement of the sentence.